# RNase E cleavage shapes the transcriptome of *Rhodobacter sphaeroides* and strongly impacts phototrophic growth

Konrad U Förstner[1,2,3,*] , Carina M Reuscher[4] , Kerstin Haberzettl[4], Lennart Weber[4], Gabriele Klug[4,*]

**Bacteria adapt to changing environmental conditions by rapid changes in their transcriptome. This is achieved not only by adjusting rates of transcription but also by processing and degradation of RNAs. We applied TIER-Seq (transiently inactivating an endoribonuclease followed by RNA-Seq) for the transcriptome-wide identification of RNase E cleavage sites and of 5′ RNA ends, which are enriched when RNase E activity is reduced in *Rhodobacter sphaeroides*. These results reveal the importance of RNase E for the maturation and turnover of mRNAs, rRNAs, and sRNAs in this guanine-cytosine-rich α-proteobacterium, some of the latter have well-described functions in the oxidative stress response. In agreement with this, a role of RNase E in the oxidative stress response is demonstrated. A remarkably strong phenotype of a mutant with reduced RNase E activity was observed regarding the formation of photosynthetic complexes and phototrophic growth, whereas there was no effect on chemotrophic growth.**

## Introduction

Most bacteria are exposed to frequent and sometimes rapid changes in their environment. Successful adaption to these changing conditions demands the ability to rapidly reprogram gene expression. Over the past years, we have learned a great deal about how bacteria sense environmental cues and use regulatory factors to switch genes on and off at the level of transcription. Rapid adjustment of the level of mRNAs by transcriptional regulation demands a quick removal of mRNAs, which requires the involvement of several ribonucleases. However, mRNA decay is not only a prerequisite for transcriptional regulation but also provides another step of gene regulation. The half-lives of mRNA can vary in response to environmental cues or in different growth phases not only because of altered levels of ribonucleases and altered interaction with sRNAs or proteins, but also

because of RNA modifications, structural changes, or cellular localization of RNAs or RNA-degrading enzymes and enzyme complexes (reviewed in Mohanty & Kushner [2016]).

In Gram-negative bacteria, the essential RNase E has a central role in RNA processing and degradation (reviewed in Mohanty & Kushner [2016]; Evguenieva-Hackenberg & Klug [2011]; Hui et al [2014]). It was initially discovered for its role in rRNA maturation (Apirion, 1978) and later turned out to affect bulk mRNA stability (Mudd et al, 1990; Vanzo et al, 1998; Bernstein et al, 2004). It also includes a membrane-binding helix (Khemici et al, 2008) and two arginine-rich RNA-binding domains (McDowall & Cohen, 1996). The *Escherichia coli* degradosome harbors the 3′–5′ exoribonuclease polynucleotide phosphorylase (PNPase), the helicase RhlB, and the glycolytic enzyme enolase in addition to RNase E (Carpousis et al, 1994; Py et al, 1996; Vanzo et al, 1998). RNase E can initiate mRNA decay by direct access or by a 5′ end-dependent pathway (reviewed in Hui et al [2014]; Fig 1A and B).

For internal cleavage, RNase E prefers single-stranded AU-rich regions (Ehretsmann et al, 1992; McDowall et al, 1994). In addition, RNase E has a strong preference for RNAs with monophosphorylated, unpaired 5′ ends (Mackie, 1998; Fig 1B). Monophosphorylated 5′ ends are generated from primary triphosphorylated 5′ ends by RNA pyrophosphohydrolase or by prior endonucleolytic cleavage of the RNA. Binding of RNase E to a 5′ monophosphorylated end promotes subsequent endonucleolytic cleavage. The 5′ fragment generated by endonucleolytic cleavage by RNase E or another endoribonuclease such as RNase III usually lacks any protective 3′ secondary structures and is, therefore, rapidly degraded by 3′–5′ exoribonucleases (Fig 1). If structural features impede the exonucleolytic decay, addition of A-rich tails by poly(A) polymerase or PNPase can promote exonucleolytic decay. The 3′ fragment generated by endonucleolytic cleavage harbors a 5′ monophosphate, which will be recognized by RNase E and promote further endonucleolytic cleavage in a 5′–3′ overall direction (reviewed in Hui et al [2014]).

Besides *E. coli* and *Bacillus subtilis*, the facultative phototrophic α-proteobacterium *R. capsulatus* is one of the best studied

[1]German National Library of Medicine—Information Center for Life Sciences, Cologne, Germany [2]Technical University of Cologne, Faculty for Information and Communication Sciences, Cologne, Germany [3]Core Unit Systems Medicine, Institutes of Molecular Infection Biology, University of Würzburg, Würzburg, Germany [4]Institut für Mikrobiologie und Molekularbiologie, Interdisciplinary Research Center for Biosystems, Universität Giessen, Giessen, Germany

Correspondence: konrad.foerstner@uni-wuerzburg.de; gabriele.klug@mikro.bio.uni-giessen.de
*Konrad U Förstner and Carina M Reuscher contributed equally to this work.

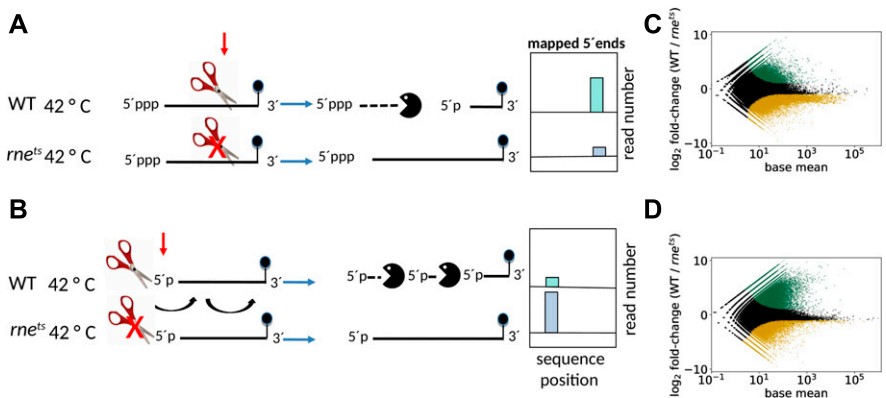

**Figure 1. Generation of stable 5′ ends in wild type and 2.4.1*rne*[E. coli(ts)] strains of *R. sphaeroides.***
**(A)** Internal cleavage by RNase E. The ends of primary transcripts are protected from degradation by a triphosphate at their 5′ end and by secondary structures (often terminators) at their 3′ end. Internal cleavage of RNase E generates an unprotected 3′ end, which allows rapid degradation by 3′–5′ exoribonucleases. The new monophosphorylated 5′ end is a new target for RNase E. The 5′ end stemming from RNase E cleavage is accumulated in the wild type at the nonpermissive temperature. (B) 5′ end–dependent degradation by RNase E. RNase E can bind to monophosphorylated 5′ ends if a stretch of unpaired nt is present and will subsequently introduce cleavages in an overall 5′–3′ direction. The 5′ monophosphate ends can stem from previous RNase E cleavage, from cleavage by other endoribonucleases, or by the action of a pyrophosphohydrolase. The 5′ monophosphate end is enriched in the *rne* mutant strain compared with the wild type at the nonpermissive temperature. Global analysis of 5′ end profiles at a permissive (32°C) (C) and a nonpermissive temperature (42°C) (D). The plots show average first-base-in-read-coverage level in wild-type samples compared with 2.4.1*rne*[E. coli(ts)] samples and the relative log₂ fold change. The x axis (base mean) represents the average, library-size normalized coverage values, whereas the y axis (fold change) represents the ratio of the normalized coverage values of the mutant in comparison with the wild type (both base-means and fold changes were calculated by DESeq2). The green dots represent sites with a significant (i.e., Benjamini–Hochberg adjusted, $P < 0.05$) enrichment in the wild type, whereas the brown dots represent sites with a significant enrichment in the mutant. Black dots represent sites without a significant enrichment.

organisms in regard to the mechanisms and regulation of mRNA degradation. α-proteobacteria comprise not only members with high metabolic versatility, such as *Rhodobacter* species, but also plant and animal symbionts and pathogens. An important role for RNase E in quorum sensing and S-adenosylmethionine homeostasis was demonstrated in *Sinorhizobium meliloti* (Baumgardt et al, 2016, 2017). Initial studies in *R. capsulatus* addressed the degradation of the polycistronic *pufQBALMX* mRNA, which encodes proteins that build the photosynthetic complexes (reviewed in Klug [1993]). Differential stabilities of the individual *puf* mRNA segments contribute to the molar ratio of light harvesting and reaction center complexes (Klug et al, 1987) and are the consequence of the distribution of RNase E cleavage sites and stabilizing RNA secondary structures (Klug, 1993). Initial cleavages of the polycistronic transcript within the *pufQ* and the *pufL* coding regions are catalyzed by RNase E, and, in the case of *pufL*, influenced by oxygen tension (Klug, 1991; Klug et al, 1992; Fritsch et al, 1995; Heck et al, 1999, 2000). As in *E. coli*, *R. capsulatus* RNase E is part of a degradosome complex (Jäger et al, 2001), which also harbors two DEAD-box RNA helicases, the transcription termination factor Rho, and minor amounts of PNPase. The composition and activity of the *R. capsulatus* degradosome vary under different oxygen concentrations (Jäger et al, 2004b). *R. capsulatus* and *R. sphaeroides* were intensely studied in regard to the redox- and light-mediated formation of photosynthetic complexes (Zeilstra-Ryalls & Kaplan, 2004) and their response to oxidative and photooxidative stress (Glaeser et al, 2011). Initial global approaches were restricted to the model organism *R. sphaeroides*, whose genome was the first to become available in this group of bacteria (Mackenzie et al, 2001; Choudhary et al, 2007). RNA-Seq analyses were performed for the first time in 2009 in *R. sphaeroides* and allowed the identification of numerous sRNAs, some of them responding to oxidative or photooxidative stress (Berghoff et al, 2009). An important function of such sRNAs in the photooxidative stress response is well documented (Mank et al, 2013; Adnan et al, 2015; Billenkamp et al, 2015; Müller et al, 2016; Peng et al, 2016). The sRNA PcrZ is a key regulator of photosynthesis gene expression and is part of a regulatory circuit that balances expression of those genes in response to changing redox conditions. It is known that trans-encoded sRNAs affect gene expression by influencing stability and/or translation of their target mRNAs (Desnoyers et al, 2013; Lalaouna et al, 2013), and the role of ribonucleases in sRNA-mediated regulation is well recognized (Viegas & Arraiano, 2008). For example, in *R. sphaeroides*, the sRNA SorX is generated by RNase E–mediated cleavage from the 3′ UTR of an mRNA (Peng et al, 2016), the sRNA SorY influences the stability of its target mRNA *takP* (Adnan et al, 2015), and the most abundant sRNA, UpsM, is processed by RNase E and derived from the 5′ UTR of the cell division gene cluster (Weber et al, 2016). Considering the acknowledged role of RNase E in several regulatory mechanisms, we performed transcriptome-wide in vivo mapping of RNase E cleavage sites (which are reduced on RNase E depletion) and of 5′ ends, which are enriched in an *rne* mutant of *R. sphaeroides*. The role of RNase E for the maturation of selected substrates was confirmed by Northern blot analyses. Furthermore, we demonstrate that RNase E can affect physiological processes differentially and has a major impact on the phototrophic growth of *R. sphaeroides*.

## Results and Discussion

### A transcriptome-wide map of RNase E cleavage sites and sites enriched in the *rne* mutant in vivo

The action of RNases is essential for bacteria to respond rapidly to changing conditions by adjusting the transcriptome. For a deeper understanding of transcriptome adjustments, we need to know the target sites of the individual RNases. Till now, a transcriptome-wide map of cleavage sites is available only for RNase E of the pathogenic γ-proteobacterium *Salmonella*. Considering our knowledge on the importance of RNA processing and degradation in the regulation of photosynthesis genes and in stress responses of *Rhodobacter*, we performed a transcriptome-wide mapping of RNase E sites in this α-proteobacterium to get insight into the global impact of this

RNase on RNA processing and decay. In contrast to a previous study in *Salmonella enterica* (Chao et al, 2017), we additionally mapped 5′ ends that are enriched in the mutant and studied their distribution.

As in other bacteria, it was not possible to delete the *rne* gene from the *R. sphaeroides* chromosome, demonstrating that RNase E is essential in this organism (data not shown). To study the relevance of RNase E cleavage for the shaping of the *R. sphaeroides* transcriptome, we replaced the *R. sphaeroides rne* gene by the *rne-3071(ts)* gene from *E. coli* strain N3431 (Weber et al, 2016). The N-terminal catalytic region shows 46% identity between the RNase E enzymes of the two species (53% identity within the S1 domain). There is no significant similarity in the C-terminal degradosome-scaffolding domain. The *R. sphaeroides* wild-type 2.4.1 and the mutant 2.4.1*rne*$^{E.coli(ts)}$ showed identical growth behavior when grown chemotrophically at 32°C (see below), demonstrating that the *E. coli* enzyme can functionally replace the *R. sphaeroides* RNase E under these conditions. Total RNA was isolated from the two strains, which were harvested in the mid-exponential growth phase at 32°C or 20 min after a shift to 42°C. Comparative RNA-Seq (three replicates each from three combined cultures) was applied to determine the 5′ ends from biological triplicates of the two strains under both growth conditions. The high correlation within the triplicates is visualized in a scatterplot of a principal component analysis (Fig S1).

The result of the enrichment analysis is visualized in Fig 1C and D by MA plots which display the average first-base-in-read-coverage level of each filtered position, and the log$_2$ fold change of the coverages of the wild-type strain libraries compared with that of the mutant strain libraries calculated by DESeq2 (Love et al, 2014). Significant depletion of 5′ ends was detected in the mutant strain compared with the wild type after the shift to 42°C. Such a depletion was observed to a lesser extent with RNA from 32°C cultures (Fig 1C and D, green spots). Differences between the two strains at 32°C can be expected because the mutant harbors an *rne* variant from *E. coli*, which may not be able to completely restore the function of the endogenous *R. sphaeroides* RNase E under all conditions. Especially, the interaction of the two RNase E variants with the subunits of the degradosome, which also varies between *E. coli* and *R. capsulatus* (Jäger et al, 2001), may be different. According to the classification of Aït-Bara & Carpousis (2015), *E. coli* RNase E belongs to type I enzymes, whereas RNase E from α-proteobacteria belongs to type II enzymes, which lack a membrane-targeting domain. As a consequence, the RNase E enzymes from the two species may also differ in cellular localization and spatial organization of the transcriptome, as this was shown for the *E. coli* and *Caulobacter crescentus* enzymes (Montero Llopis et al, 2010; Bayas et al, 2018). Furthermore, it is conceivable that the RNase E activity of strain 2.4.1*rne*$^{E.coli(ts)}$ is already slightly impeded at 32°C. At 28°C, which is often chosen as the permissive temperature for the RNase E variant in *E. coli*, growth of *R. sphaeroides* is too slow to compare it with growth at 42°C.

Interestingly, the MA plots revealed a similar pattern for 5′ ends enriched in the mutant at 32°C and at 42°C (Fig 1C and D, orange spots). At 42°C, ~16,000 5′ ends with such an enrichment pattern were detected in the mutant and at 32°C, even ~19,000 5′ ends were enriched. Further analyses revealed only a small overlap between the 5′ ends enriched in the mutant at 32°C or at 42°C (about 2,500 5′

ends). One has to consider that structural features of RNA also affect its degradation and that the temperature shift to 42°C may have affected the formation of secondary structures. This may be a reason for the different sets of enriched sites at the two temperatures.

Because the primary 5′ end of a transcript is rapidly attacked by RNase E, if the triphosphate is converted into monophosphate by the action of pyrophosphohydrolase, then stabilization in the RNase E mutant can be expected (Fig 1B). Other 5′ ends which accumulate in the RNase E mutant can be expected to arise from primary endonucleolytic cleavage by other endoribonucleases, which also results in a monophosphorylated 3′ fragment. In addition to RNase E, endoribonuclease G, which has strong homology to the N-terminal part of RNase E, and the double-strand–specific RNase III participate in endonucleolytic cleavage of RNA in Gram-negative bacteria (Evgenieva-Hackenberg & Klug, 2009). *R. sphaeroides* also harbors an RNase J enzyme (Rische-Grahl et al, 2014), which was shown to exhibit endonucleolytic activity in *B. subtilis* (Even et al, 2005).

For further analyses, we included all positions with a normalized first base-in-read coverage fold change of equal or higher than two between the two strains at 42°C and an adjusted *P*-value of 0.05 (calculated by DESeq2, marked in green or brown in the MA plot in Fig 1C and D). This leads to about 41,000 5′ ends, which were depleted in the mutant at the nonpermissive temperature. We consider these 5′ ends as *bona fide* RNase E cleavage sites (Fig 1A). On the other hand, about 16,000 5′ ends were depleted in the wild-type strain at the nonpermissive temperature.

5′ ends enriched in the wild type and those enriched in the mutant are often detected at neighboring nucleotides. Because endoribonucleases have little sequence specificity, the cut may not be at a distinct position, especially not within homonucleotide stretches. Because of this, we consider 5′ ends, which are directly adjacent to each other (in a proximity of 3 nt based on *bedtools cluster* subcommand; https://bedtools.readthedocs.io/en/latest/content/tools/cluster.html), to belong to the same cleavage site and considered the center position of the cluster in the downstream analysis.

For further analyses of the cleavage site distribution, we therefore performed a clustering of 5′ ends and clustering of the sites enriched in the mutant (from now on, enriched sites). This reduced the 41,000 5′ ends depleted in the mutant to 23,000 cleavage sites and the 5′ ends enriched in the mutant from 16,000 to 10,000 enriched sites. Fig 2 shows the distribution of cleavage sites (Fig 2A) and the distribution of enriched sites (Fig 2B) among different types of RNAs. Most of the cleavage sites (~18,000, 87%) and enriched sites (~8,000, 87%) are found in the coding sequence of mRNAs, which is not surprising because they represent by far the majority of RNA reads. The box plots (Fig 2C and D) show a big variation in the cleavage site density (cleavage sites per kilobase) and density of enriched sites within the coding region. Considering noncoding RNA regions, most cleavage sites are found within the 5′ UTRs of mRNAs (~2,000, ~10%), which may reflect the regulatory role that is often assigned to 5′ UTRs. Only about 2% of the cleavage sites mapped to 3′ UTRs; similar ratios were described by Chao et al (2017) in the related study on *S. enterica* RNase E. Although the distribution of cleavage sites is mostly similar to that of the

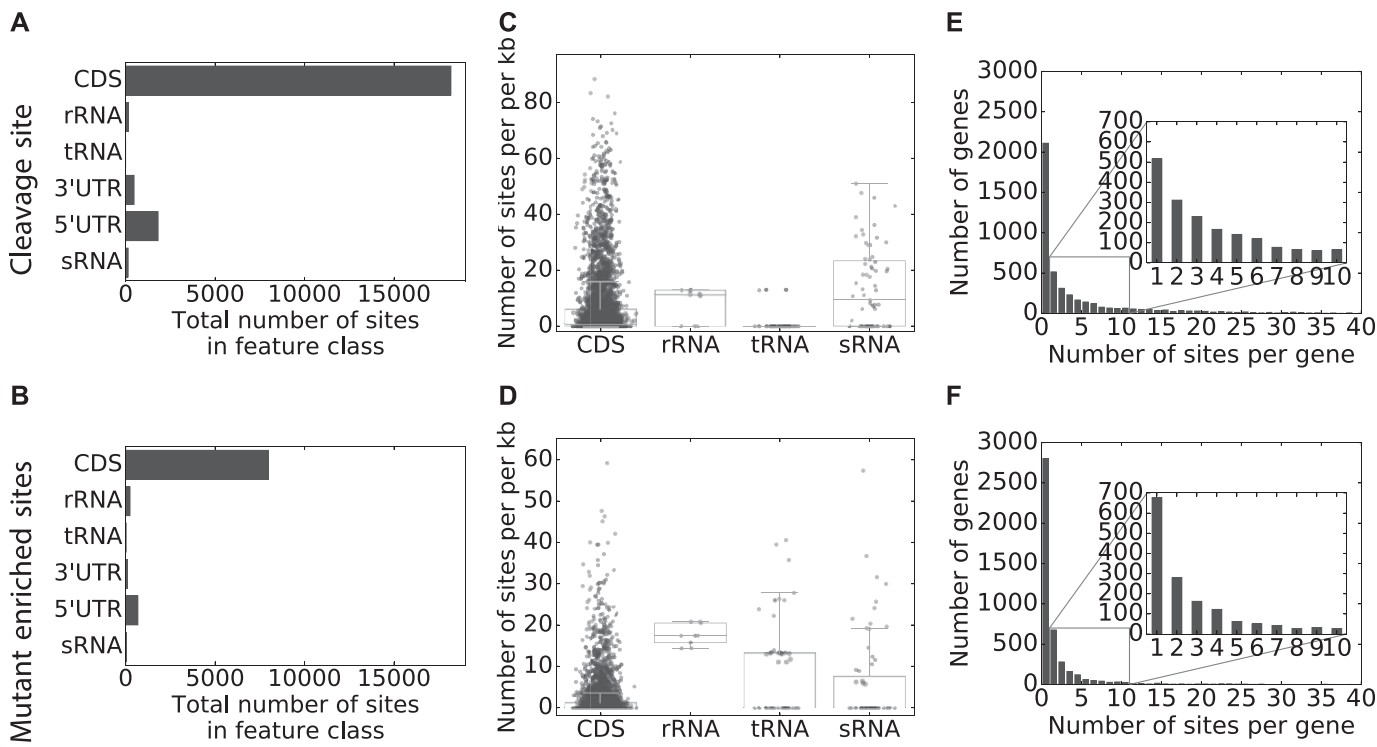

**Figure 2. Statistics of cleavage sites and mutant-enriched sites.**
**(A)** Sum of all RNase E cleavage sites mapped to different RNA categories. **(B)** Sum of all enriched sites mapped to different RNA categories. **(C)** Box plot of the cleavage site density for the individual genes per kilobase sorted by gene type. **(D)** Box plot of mutant enriched site density for the individual genes per kilobase sorted by gene type. **(E)** Histogram of the distribution of RNase E cleavage site density in genes per kilobase. **(F)** Histogram of the distribution of mutant enriched site density in genes per kilobase.

enriched sites (Fig 1B), it is remarkable that a higher percentage of enriched sites (~3%) compared with cleavage sites (~1%) occurs within rRNAs. The box plots show that the density of enriched sites within the rRNA (up to 22 sites per kilobase) is higher than the cleavage site density (Fig 2C and D). This suggests that many primary cleavages within rRNA are catalyzed by other RNases and the monophosphorylated cleavage products are stabilized in the RNase E mutant. Important roles in rRNA processing have been established for RNase III and for RNase J in *Rhodobacter* (Evguenieva-Hackenberg & Klug, 2000; Rische & Klug, 2012). Fig 2E and F illustrates the large variation of cleavage sites or enriched sites per gene. Remarkably, a high number of genes lack any cleavage sites (about 2,100) or enriched sites (about 2,800).

We performed a motif analysis for the depleted and enriched 5′ ends as well as for the cleavage sites and enriched sites using MEME (Bailey et al, 2009). This sequence motif analysis revealed that ~9,000 of all 41,000 5′ ends depleted in the mutant and ~7,000 of the 23,000 clustered cleavage sites clearly have the nucleotides AU enriched around the cleavage position (Fig 3A).

Based on an analysis of selected substrates, preference of RNase E cleavage for AU-rich sequences has been reported previously (Ehretsmann et al, 1992; McDowall et al, 1994). Global mapping of RNase E cleavage sites in *E. coli* identified AU preference at position +1 and a strong preference for U at position +2 (Chao et al, 2017). It should be noted that *R. sphaeroides* and *R. capsulatus*, in contrast to *E. coli* (50.8% GC), have a high GC content of 68.8 and 66.6%, respectively. Nevertheless, analysis of selected cleavage

sites within the *puf* operon of *R. capsulatus* that encodes protein components of the photosynthetic complexes also revealed a preference for AU-rich target sequences (Fritsch et al, 1995; Heck et al, 1999, 2000). A preference for AU-rich targets was also detected in our global analysis (Fig 3A). For the sites enriched in our global analysis, MEME found the pattern UCGA (Fig 3B), but only for a small fraction of sites (629 enriched 5′ ends/439 clustered enriched sites).

To investigate the connection between cleavage sites and coding regions, the frequency of cleavage sites in dependency of their relative position of start and stop codons was analyzed (Fig S2). Although RNase E cleavage sites were observed to occur frequently in close proximity to stop codons in *Salmonella* (Chao et al, 2017), we found a relatively high incidence of cleavage sites around start codons.

### The effect of RNase E on maturation/abundance of selected substrates

RNase E was shown to be involved in the maturation and processing of many sRNAs in enterobacteria (Massé et al, 2003; Lalaouna et al, 2013). Our analysis confirms a role of RNase E also in the maturation/processing of sRNAs in the GC-rich α-proteobacterium *R. sphaeroides*. We have analyzed the effect of RNase E on the maturation of sRNAs that are derived from the 5′ or 3′ UTR of mRNAs or that have their own promoter. Previous studies indicated that processing of the 3′ UTR–derived sRNA SorX and processing of the 5′ UTR–derived sRNA UpsM require RNase E activity (Peng et al, 2016; Weber et al, 2016). Fig 4A and B shows the results of the TIER-Seq

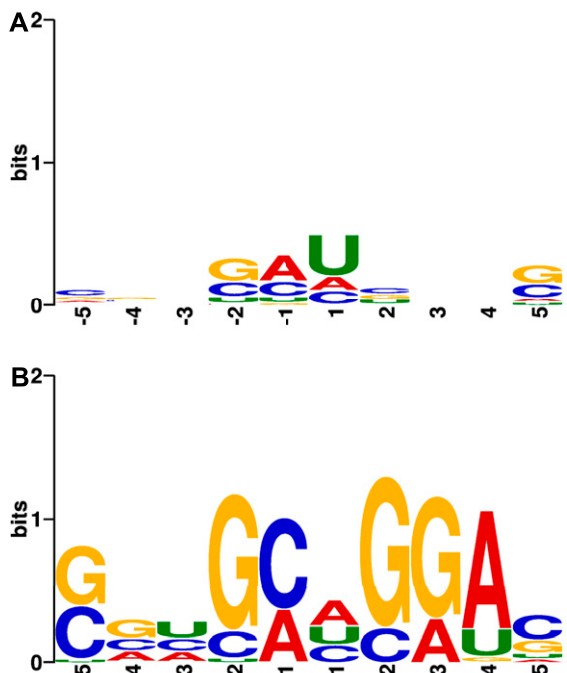

**Figure 3. Consensus sequence motifs.**
**(A)** Sequence motif of RNase E cleavage sites based on alignment of all mapped cleavage sites (with one base of shifting allowed). **(B)** Sequence motif around 5' ends enriched in the *rne* mutant based on the alignment of all mapped cleavage sites (with one base of shifting allowed).

analysis together with the Northern blots for these two sRNAs. In both cases, the *R. sphaeroides* strain harboring the *rne*[E. coli(ts)] gene from *E. coli* is already impeded in sRNA processing at 32°C, demonstrating that the *E. coli* protein cannot fully replace the *R. sphaeroides* protein at this temperature.

The coverage plot in Fig 4A shows the RNase E cleavage site at the 5' end of the 130-nt processing product of UpsM (total 206 nt) in the TIER-Seq analysis (genome position: 694,720). UpsM is transcribed from an upstream promoter, and read-through past the UpsM terminator allows transcription of the 3' localized cell division genes (Weber et al, 2016). In agreement with RNase E–dependent processing, the 130-nt fragment is much less abundant in the mutant strain. The shift to 42°C leads to reduced levels of the UpsM 206-nt transcript in the wild type, which may be due to reduced promoter activity or decreased transcript stability. Shorter RNA half-lives at elevated temperatures are frequently reported and also observed in *R. sphaeroides* (Jäger et al, 2004a, b, c). The shift to 42°C leads to a complete lack of the 130-nt-long UpsM transcript in the mutant, supporting the RNase E–dependent processing.

Like UpsM, the sRNA RSs1624 is derived from the 5' UTR of an mRNA that encodes a hypothetical protein (RSP_6083). The main 5' end (genome position: 1,724,505) coincides with the TSS and is clearly enriched in the mutant, indicating reduced 5' end–dependent decay upon reduced RNase E activity. This is reflected by higher abundance of RSs1624 in the mutant strain as detected by Northern blot (Fig S3A).

The SorX sRNA is processed from the 3' UTR of a transcript encoding RSP_0847, an OmpR-like protein (Peng et al, 2016). Fig 4B

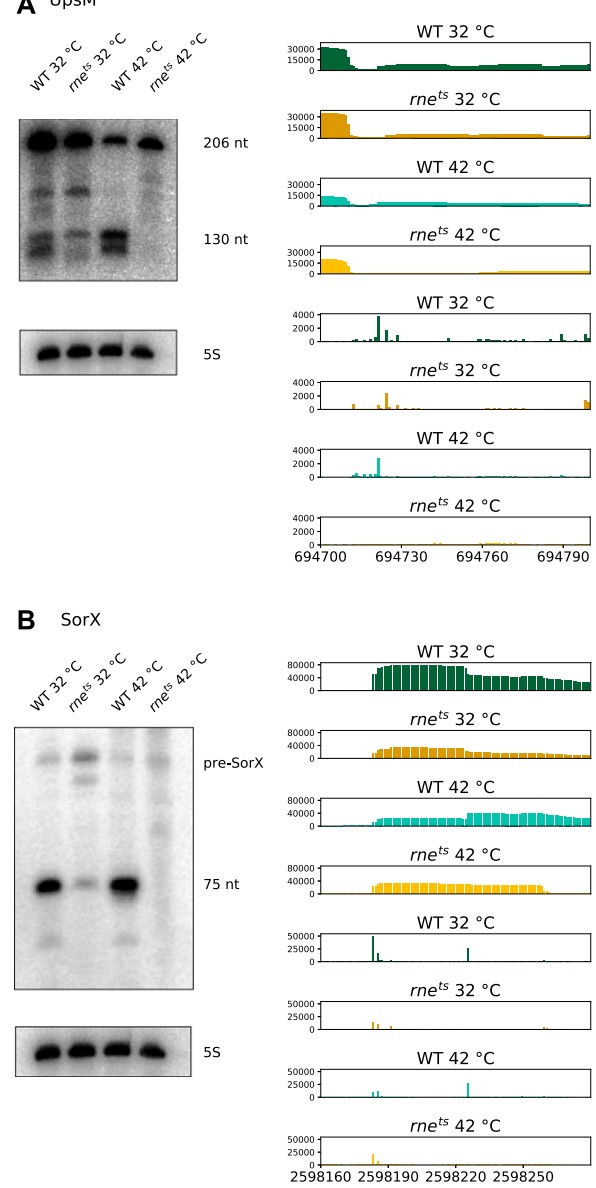

**Figure 4. Northern blot analysis of the sRNAs UpsM (A) and SorX (B) in the wild type and the 2.4.1*rne*[E. coli(ts)] mutant.**
For RNA isolation, the strains were cultivated under microaerobic conditions at 32°C and shifted for 20 min to 42°C. 10 µg of total RNA was blotted. 5S rRNA was used as a loading control. Read coverage for UpsM (A) and SorX (B) in the wild type (green/light green) and 2.4.1*rne*[E. coli(ts)] (brown/orange) at 32°C (dark colors) and 42°C (bright colors). Upper panels show the coverage of all reads, whereas the lower panels show the 5' ends with single-nucleotide resolution.

depicts the RNase E cleavage site at the 5' end of the 75-nt SorX RNA (genome position: 2,598,224). The 5' end of the 116-nt pre-SorX RNA (genome position: 2,598,182) is enriched in the RNase E mutant at 42°C, implying that primary processing steps from the RSP_0847 transcript are catalyzed by enzymes different from RNase E, but that the lack of RNase E reduces the 5' end–dependent pathway of degradation and stabilizes pre-SorX as also indicated by the Northern blot. Another sRNA is processed from the 3' UTR of RSP_1771, which is annotated as citrate lyase subunit. The function

of this sRNA is not known. The TIER-Seq results (Fig S3B) reveal an RNase E cleavage site at the 5′ end of the sRNA (genome position: 355,667), and the Northern blot confirms that reduced RNase E activity results in reduced levels of the sRNA. Again, the temperature increase per se reduces the levels of the sRNA.

Like SorX (Peng et al, 2016), the sRNA SorY has a role in the oxidative stress response (Adnan et al, 2015). Unlike the other sRNAs depicted here, SorY has its own promoter. The main 5′ end of SorY (genome position: 1,632,642) maps to the TSS and is clearly enriched in the mutant strain (Fig S3C) in agreement with the 5′ end–dependent degradation of primary transcripts by RNase E (Fig 1B). Another sRNA with known function in *R. sphaeroides* is PcrZ (136 nt), which is involved in the regulation of photosynthesis gene expression as part of a regulator circuit (Mank et al, 2013). It is transcribed from its own promoter and further processed to stable sRNAs of about 55 nt, which are not functional (Mank et al, 2012). The reads mapping to the main 5′ end of PcrZ (position: 2,565,820) or within the PcrZ sequence did not show significant differences between the wild-type *R. sphaeroides* or the *R. sphaeroides* strain harboring the *rne^{E. coli(ts)}* gene from *E. coli* (Fig S3D). In agreement with this, the corresponding Northern blot indicates that the amount of the 136-nt transcript does not significantly vary in the two strains at the permissive or nonpermissive temperature (Fig S3D). The TIER-Seq analysis, however, identified an RNase E cleavage site at the 5′ end of the processing products (genome position: 2,565,914), but only a slightly decreased amount in the mutant is observed on Northern blot. Indeed, both the 136-nt PcrZ transcript and the processing products have a very long half-life and no decay is observed within 160 min after the addition of rifampicin (data not shown). Inactivation of RNase E by a 20-min shift to elevated temperature will, therefore, have only minor effects on the amount of PcrZ transcripts. However, altered RNase E activity over longer time periods may well impact PcrZ processing and, thus, contribute to the effect of RNase E on the formation of photosynthetic complexes. Thus, the TIER-Seq analysis provides important information on the involvement of RNase E in processing of selected RNAs that cannot be obtained by just performing Northern blots with RNA from the two strains grown at different temperatures.

The effects of RNase E as predicted by the TIER-Seq and RNA-Seq analyses can also be tested on Northern blots in case of small mRNAs. Fig S3E shows the results for a 300-nt-long RNA encoding a small hypothetical protein (RSP_7527), which is induced at elevated temperatures. The 5′ end of this RNA (genome position: 1,289,874; Fig S3E, part 1) is enriched in the *rne* mutant and, consequently, levels of the 300-nt transcript in the mutant are higher at 42°C. This is another example for stabilization of an RNA, which is degraded by RNase E by the 5′ end–dependent pathway in the wild type (Fig 1B). An sRNA of about 70 nt is processed from the 3′ UTR of this transcript by RNase E, which leads to higher levels at 32°C in the wild type (5′ end of the processing product at genome position: 1,290,108; Fig S3E, part 2). Because at 42°C the total transcript level is higher in the mutant, the decrease of the small processing product is not obvious on the Northern blot. Quantification of the band intensity revealed that for the wild type, the processed band has 52% of the intensity of the primary transcript, whereas these are only 28% in the mutant, which is in agreement with reduced

processing and reduced levels of the processing product, when RNase E activity is decreased.

For the mRNA encoding another small hypothetical protein (RSP_7517), the TSS could not be unambiguously identified. The main 5′ end (genome position: 798,199) is enriched in the *rne* mutant and leads to a band of about 240 nt on Northern blots, which is clearly more abundant in the mutant strains (Fig S3F). We conclude that this RNA species is increased in abundance because of weaker 5′ end–dependent degradation by RNase E (Fig 1B). Another 5′ end more upstream shows no clear RNase E–dependent changes (not shown) and gives rise to a band of about 265 nt (Fig S3F).

Our TIER-Seq analysis, thus, confirms the importance of RNase E for the maturation of 5′ UTR– or 3′ UTR–derived sRNAs and also for the processing of sRNAs. In contrast to the previous study in *S. enterica* (Chao et al, 2017), our analysis also considers the 5′ ends enriched in the *rne* mutant. This approach allows to follow the accumulation of 5′ ends of primary transcripts or processing products due to reduced 5′ end–dependent RNase E activity and can retrieve additional information from the TIER-Seq data set.

## Influence of RNase E on the formation of photosynthetic complexes and oxidative stress resistance

Considering the established role of RNase E in the regulation of photosynthesis gene expression in *R. capsulatus* (Klug, 1993) and in the generation or processing of sRNAs with a regulatory role in stress responses in *R. sphaeroides* (Peng et al, 2016; Weber et al, 2016), it was not surprising to observe clear phenotypes for the strain expressing the temperature-sensitive RNase E. Even at the normal growth temperature of 32°C, the mutant strain showed significantly lighter color than the wild type, indicating lower levels of photosynthetic complexes. It was shown in a previous study (Weber et al, 2016) and in the analyses of this study that at normal growth temperature, the *rne-3071* gene from *E. coli* leads to some differences in processing when compared with the wild-type situation.

Altered levels of photosynthetic complexes were confirmed by determining the bacteriochlorophyll (BChl) content (Fig S4) and monitoring the absorption spectra (Fig 5A). The BChl amount was about threefold higher in the wild-type strain than in the mutant under microaerobic (chemotrophic growth) or phototrophic growth conditions at 32°C. This corresponds to the observation that photosynthetic growth of the mutant is strongly retarded (Fig 5B). Remarkably, both strains showed identical growth behavior when grown chemotrophically. This finding underlines the important physiological role of RNase E and further demonstrates that RNase E can have a selective impact on specific physiological processes.

RNA processing has an important role in the expression of photosynthesis genes in *R. capsulatus* (reviewed in Klug [1993]; Rauhut & Klug [1999]). The polycistronic *pufQBALMX* operon (Fig S5) encodes pigment-binding proteins (*pufB* and *pufA* for light-harvesting complex I [LHI] and *pufL* and *pufM* for the reaction center), proteins involved in the assembly of the complexes (*pufX*) and another one affecting early BChl synthesis (*pufQ*) (Fidai et al, 1995). Segmental differences in *puf* mRNA stability contribute to the stoichiometry of LHI to reaction center complexes (Klug et al, 1987).

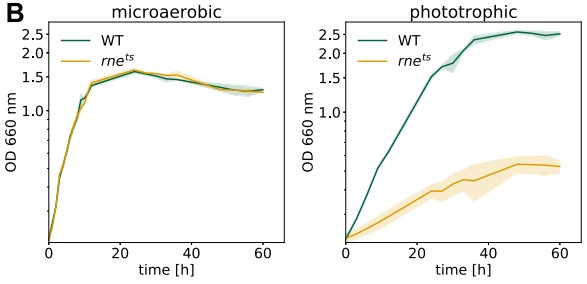

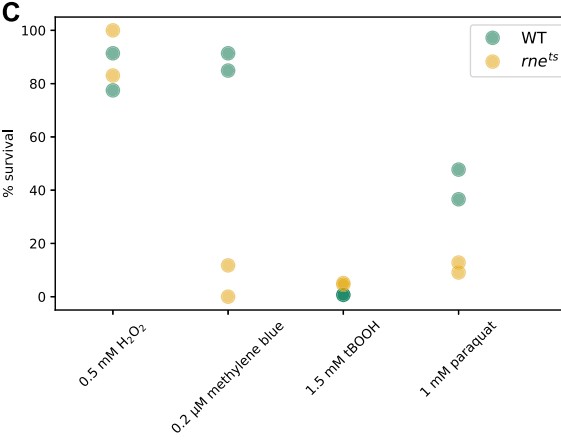

**Figure 5.** **(A)** Absorption spectra of cell-free extracts. Spectra were obtained from cultures grown either under microaerobic or phototrophic conditions to an $OD_{660}$ of 0.8. The absorption between 450 and 600 nm is caused by carotenoids, whereas the peaks at 800 and 850 nm are caused by the BChls bound to LHII (B800-850) and LHI (B870). The spectra are each for one representative culture. **(B)** Cultures of the wild type of *R. sphaeroides* (green) and 2.4.1$rne^{E.\ coli(ts)}$ (orange) mutant were cultivated at 32°C under microaerobic conditions in the dark or in the presence of light (60 W) under anaerobic conditions. Optical density was measured at 660 nm. The curves represent the mean value of biological triplicates. **(C)** Survival assay of the wild type of *R. sphaeroides* and 2.4.1$rne^{E.\ coli(ts)}$ mutant. The strains were cultivated at 32°C under microaerobic conditions in the dark to an $OD_{660}$ of 0.4 and shifted to 42°C for 1 h either under no-stress conditions for control or in the presence of either 0.5 mM $H_2O_2$, 1.5 mM tBOOH, or 1 mM paraquat. For photooxidative stress, bacteria were cultivated at 32°C under aerobic conditions in the dark and in the presence of 0.2 $\mu$M methylene blue until reaching an $OD_{660}$ of 0.4. The cells were then shifted to 42°C in the presence of light (800 W/m$^2$) for 1 h, whereas the controls were cultivated in the dark. Survival is displayed as the percentage of colony numbers forming in serial dilutions of the stressed bacterial cultures relative to the control culture (referred to as 100% survival). The average of two measurements of two independent cultures is shown.

Rate-limiting cleavage of the *pufQBALMX* mRNA is initiated by RNase E at an AU-rich sequence within the *pufQ*-coding region (Heck et al, 2000), which leads to a more stable *pufBALMX* mRNA. Again, RNase E initiates further processing by cleaving an AU-rich site in the 5′ region of the *pufL*-coding sequence (Fritsch et al, 1995; Heck et al, 2000). A similar processing pattern as in *R. capsulatus* is observed for the *R. sphaeroides puf* mRNA (Li et al, 2004), and our TIER-Seq approach detected RNase E cleavage sites also at an AU-rich sequence within the 5′ region of *R. sphaeroides pufL* (position: 1,982,393) and in the intergenic region between *pufA* and *pufL* (position: 1,982,467; Fig S5). However, there are more RNase E cleavage sites and enriched sites distributed within the *puf* operon and other photosynthesis genes (e.g., for pigment synthesis), and further experimental analyses will be required to identify primary sites that limit the overall rate of processing. As in the *puf* operon, most other photosynthesis genes are organized in long polycistronic operons. Several mRNAs (e.g., *pucB*, *bchC*, *bchF*, *crtE*, and *crtA*; *puc* genes encode proteins of the LHII complex, *bch* genes code for enzymes for BChl synthesis, and *crt* genes code for enzymes for carotenoid synthesis) have a 5′ end at the predicted TSS, which is enriched in the mutant strain at the nonpermissive temperature, most likely because of the lack of 5′-dependent RNase E activity. Some mRNAs encoding important regulators of photosynthesis genes (*appA*, *ppaA*, and *fnrL*) harbor several RNase E cleavage sites and enriched sites, indicating that RNase E also acts on the formation of photosynthetic complexes by affecting the levels of such protein regulators. Nevertheless, the strong effect of RNase E on phototrophic growth cannot be deduced from our global mapping of RNase E–dependent 5′ ends. There is no particular high or low density of RNase E cleavage sites or enriched sites in photosynthesis genes. Presently, it remains elusive which particular RNase E–dependent processes have a major influence on physiological processes. It is likely that RNase E–dependent cleavage in the 5′ region of *pufL* and processing of the sRNA PcrZ contribute to the impeded formation of photosynthetic complexes in the mutant.

We already demonstrated the involvement of RNase E in the maturation of the sRNA SorX, which is derived from the 3′ UTR of gene RSP_0847 encoding an OmpR family protein. SorX has a role in the oxidative stress response (Peng et al, 2016) and the transcriptome data also identify cleavage sites in other genes, which are known or predicted to be involved in the oxidative stress response. Remarkably, the mRNAs for the alternative sigma factors RpoE (RSP_1092), RpoHI (RSP_2410), and RpoHII (RSP_0601) exhibit a high density of RNase E cleavage sites (62, 64, and 30 per kb, respectively). All these sigma factors have important roles in the response to singlet oxygen (Anthony et al, 2005; Glaeser et al, 2007; Nuss et al, 2009, 2010) and RpoHI and RpoHII also activate genes in response to other stresses such as heat or stationary phase (Adnan et al, 2015; Billenkamp et al, 2015). For these reasons, we also analyzed the role of RNase E in the resistance to oxidative stress. Zone of inhibition assays were performed for the wild type and the RNase E mutant at 32°C (Fig S6). At 42°C, no clear zones of inhibition could be obtained. We did not observe an effect of RNase E on the resistance to hydrogen peroxide or to organic hydroperoxides (tBOOH). The mutant was slightly more resistant to the superoxide generating paraquat and slightly less resistant against singlet

oxygen, which is generated in the presence of methylene blue and light. Fig 5C shows the results for survival assays of both strains at 42°C performed in liquid cultures. For hydrogen peroxide and tBOOH, only minor or even no differences in survival were observed. However, the mutant strain showed significantly decreased survival rates under singlet oxygen (methylene blue) and superoxide (paraquat) stress. This demonstrates that RNase E also affects other important physiological functions besides the formation of photosynthetic complexes and photosynthesis.

Our findings underline the central role of RNase E in RNA metabolism, despite the presence of several enzymes with ribonucleolytic activity in bacteria, and emphasize the importance of RNA processing and decay on regulation of gene expression, an aspect that is still neglected in many studies focusing on gene regulation. Fig 6 summarizes these findings in a schematic model. Although RNase E is not the only endoribonuclease in bacteria and also exoribonucleases contribute to RNA processing and degradation, reduced RNase E activity has a clear impact on the transcriptome of *R. sphaeroides*. It is highly likely that the alterations at the transcriptome level also affect the proteome. Although reduced RNase E activity has no significant effect on growth under chemotrophic conditions, phototrophic growth is strongly impeded, demonstrating that RNase E can have a selective impact on specific physiological processes.

## Materials and Methods

### Strains and growth conditions

We compared the wild-type strain *R. sphaeroides* 2.4.1 (van Niel, 1944) with strain 2.4.1*rne*[E. coli(ts)], which has the *R. sphaeroides rne* gene replaced by the temperature-sensitive *E. coli rne-3071 (ts)* allele derived from the strain *E. coli* N3431 (Apirion, 1978; Goldblum & Apririon, 1981). Construction of this strain is described in detail

elsewhere (Weber et al, 2016). All strains were grown in a malate minimal medium (Remes et al, 2014) under chemotrophic, micro-aerobic conditions (25 $\mu$M dissolved oxygen) or under phototrophic conditions (anaerobic conditions in the presence 40 W of white light) at 32°C or 42°C.

### Determination of BChl and spectral analysis

Photopigments were extracted with acetone–methanol (7:2 vol:vol) from cell pellets, and the BChl concentration was calculated by using an extinction coefficient of 76 mM$^{-1}$·cm$^{-1}$ at 770 nm. For spectral analyses, exponential phase cells were collected, resuspended in ICM buffer (10 mM $KH_2PO_4$/$K_2HPO_4$ and 1 mM EDTA [pH 7.2]), and disrupted by sonication (power MS 72/D, cycle 70%, 5 × 30 s). The supernatants were transferred into 1.5 ml tubes and cell-free crude extracts were centrifuged in a microcentrifuge at 13,000 rpm for 20 min at 4°C to remove cells and cell debris. The protein concentration of the supernatants was determined by a Bradford assay. An amount of 100 $\mu$g of protein in a total volume of 400 $\mu$l of ICM buffer was recorded on a SPECORD 50 spectral photometer (Analytik Jena AG).

### Zone of inhibition and survival assays

Zone of inhibition assays were carried out as described previously (Li et al, 2003). Plates were incubated for 3 d at 32°C, and the diameter of the zone of inhibition indicating the sensitivity of the cells against the agent was measured. For determination of survival rates, *R. sphaeroides* cultures were grown in the presence of oxidative stress agents for 1 h at 42°C and dilutions were plated. The number of colonies of a control culture grown without the addition of any oxidative stress agents was defined as 100% survival. The percentage of colonies grown from the treated cultures (percent survival) was referred to the 100% percent of the control culture.

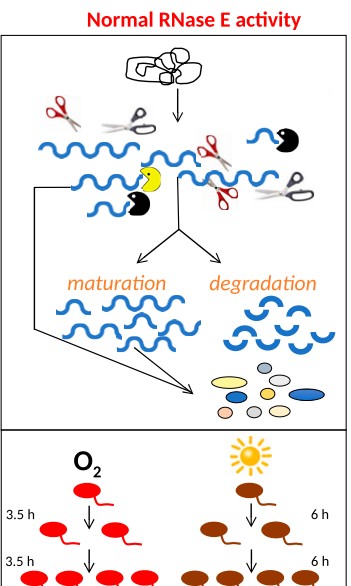
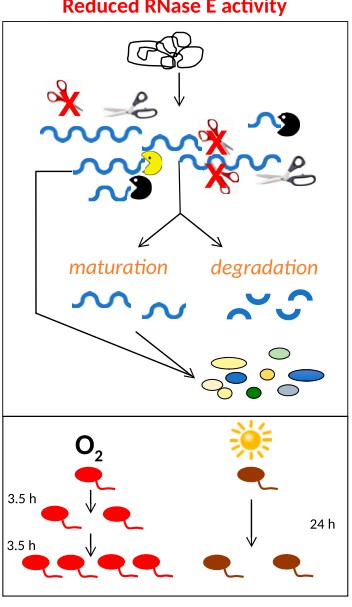

**Figure 6. Schematic overview of the importance of RNase E for *R. sphaeroides*.**
Although RNase E is only one of several RNases acting on mRNAs and sRNAs, it significantly affects the transcriptome of *R. sphaeroides*, which most likely also affects the proteome. These effects on RNA processing/degradation strongly impact phototrophic growth, whereas no significant effect on chemotrophic growth is observed.

### RNA isolation and Northern blot analysis

Total RNA was isolated by the hot phenol method (Janzon et al, 1986). Remaining DNA was removed by treating the samples with 6 U of DNase I (#18047019; Invitrogen) per 1 µg of RNA. Absence of DNA contamination was confirmed by PCR against the *gloB* gene (RSP_0799). Northern blot analysis was performed as described earlier (Berghoff et al, 2009). Oligodeoxynucleotides used for hybridization are listed in Table S1. $\alpha$-$^{32}$[P]-ATP (SRP-301; Hartmann Analytic) and T4 polynucleotide kinase (#EK0031; Fermentas) were used for the end-labeling reaction. For detection of SorX, a PCR product (primers listed in Table S1) was labeled with $\alpha$-[$^{32}$P]-dCTP (SRP-205; Hartmann Analytic) using the Prime-a-Gene Labeling System (U1100; Promega) as described in the manufacturer's manual. Source data for the Northern blots are provided in the Supplementary Information.

### Library construction and RNA sequencing

Data are based on triplicates; RNA for each triplicate stems from three independent cultures. The TIER-Seq data discussed in this publication have been deposited in NCBI's Gene Expression Omnibus (Edgar et al, 2002) and are accessible through the GEO Series accession number GSE104278.

### Bioinformatical analysis

The tool cutadapt (version 1.10) was used to remove the adapter sequences from the reads and to perform a qualitative trimming for the previously published RNA-Seq (NCBI GEO accession number: GSE71844) and the TIER-Seq data. Further read filtering by length, mapping to the reference genome (NC_007493.2, NC_007494.2, NC_009007.1, NC_007488.2, NC_007489.1, NC_007490.2, and NC_009008.1), and nucleotide-wise coverage calculation was done with READemption (version 0.4.3; Förstner et al, 2014) and segemehl (version 0.2.0; Hoffmann et al, 2009, 2014). The calling of transcription start sites and further genomic features was carried out based on the dRNA-Seq coverage files in wiggle format using ANNOgesic (version 0.5.6; Yu et al, unpublished). For the prediction of cleavage sites, the workflow described by Chao et al (2017) was adapted. Nucleotides that do not have at least in one of the libraries a first-base-in-read coverage of 10 or more were removed. The first-base-in-read-coverage values of the remaining nucleotide positions were used as input for enrichment analysis with DESeq2 (version 1.16.1; Love et al, 2014). Positions with a fold change equal or higher than two in the wild type in comparison with the mutant and an adjusted *P*-value (Benjamini–Hochberg correction) below 0.05 were considered as *bona fide* cleavage sites. Candidates at a proximity of 3 nt were merged to single cleavage sites using *bedtools* (version 2.25.0, Quinlan & Hall, 2010) subcommand cluster. The number of cleavage sites that fall into the different annotation features was quantified with *bedtool'* subcommand *intersect* and densities as sites per kilobase were calculated and plotted. For the detection of motifs, the sequences 5 nt up- and downstream of cleavage sites and enrichment sites were extracted and analyzed with MEME (version 4.11.2; Bailey et al, 2009). The whole workflow including all custom-made programs needed to reproduce the analysis is deposited at Zenodo (https://doi.org/10.5281/zenodo.824921).

## Supplementary Information

## Acknowledgements

We thank Gisela Storz for feedback regarding the manuscript and Matthew McIntosh for corrections in the manuscript. The work was funded by German Research Foundation (DFG) (Kl563/28). This publication was funded by DFG and the University of Würzburg in the funding programme Open Access Publishing.

### Author Contributions

KU Förstner: conceptualization, data curation, software, formal analysis, investigation, visualization, methodology, and writing—original draft, review, and editing.
CM Reuscher: data curation, formal analysis, validation, investigation, visualization, methodology, and writing—review and editing.
K Haberzett: investigation.
L Weber: investigation.
G Klug: conceptualization, data curation, formal analysis, supervision, funding acquisition, project administration, and writing—original draft, review, and editing.

### Conflict of Interest Statement

The authors declare that they have no conflict of interest.

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
