## [Reviewer comments · Life Science Alliance]

RNase E cleavage shapes the transcriptome of *R. sphaeroides* and strongly impacts phototrophic growth

Konrad Förstner, Carina Reuscher, Kerstin Haberzett, Lennart Weber, and Gabriele Klug
DOI: 10.26508/lsa.201800080

Review timeline:

Submission Date:	7 March 2018
Editorial Decision:	28 May 2018
Revision Received:	8 July 2018
Editorial Decision:	10 July 2018
Revision Received:	13 July 2018
Accepted:	13 July 2018

Report:

(Note: Letters and reports are not edited. The original formatting of letters and referee reports may not be reflected in this compilation.)

May 28, 2018

Re: Life Science Alliance manuscript #LSA-2018-00080-T

Dr. Konrad Ulrich Förstner
Core Unit Systems Medicine, Universität Würzburg
Josef-Schneider-Str. 2/D15
Wuerzburg D-97080
Germany

Dear Dr. Förstner,

Thank you for submitting your manuscript entitled "RNase E cleavage shapes the transcriptome of *R. sphaeroides* and strongly impacts phototrophic growth" to Life Science Alliance. The manuscript was assessed by expert reviewers, whose comments are appended to this letter. We invite you to submit a revision if you can address the reviewers' key concerns, as outlined here.

As you will see, the reviewers appreciate your analyses and provide constructive input on how to further strengthen your manuscript. They request mostly a careful revision of the text and figure display, and I would thus like to ask you to follow their advise and provide such a revised version. Please also add further evidence for the preference for AU rich target sites (reviewer #1, point 1). The potential lack of physiological significance of the dataset in light of using a type I RNase E for analyses in *R. sphaeroides* also needs to be clearly mentioned (reviewer #3).

-- High-resolution figure, supplementary figure and video files uploaded as individual files: See our detailed guidelines for preparing your production-ready images, <http://life-science-alliance.org/authorguide>

-- Summary blurb (enter in submission system): A short text summarizing in a single sentence the study (max. 200 characters including spaces). This text is used in conjunction with the titles of papers, hence should be informative and complementary to the title and running title. It should

describe the context and significance of the findings for a general readership; it should be written in the present tense and refer to the work in the third person. Author names should not be mentioned.

B. MANUSCRIPT ORGANIZATION AND FORMATTING:

Full guidelines are available on our Instructions for Authors page, <http://life-science-alliance.org/authorguide>

Thank you for this interesting contribution to Life Science Alliance. We are looking forward to receiving your revised manuscript.

Sincerely,

Reviewer #1 (Comments to the Authors (Required)):

The authors globally identified the RNase E cleavage sites and 5' ends in *Rhodobacter sphaeroides* by applying the TIER-seq method. In this method the endoribonucleases are inactivated for a short time and this is followed by RNA-seq. The results demonstrate the importance of RNase E to

different aspects in the life of *R. sphaeroides* such as: maturation and turn-over of RNAs or coping with oxidative stress. The paper is well-written and is easy to read. It also provides an important resource for researchers studying *Rhodobacter sphaeroides*. However, the paper could be improved by the following:

1. The authors state that "RNase E shows a preference for AU rich target sites". The only support they provide for this statement is a weak motif shown in Figure 3(a). The authors should consider additional experiments to support this statement or to better address this issue. For example, replacing the AU in one of the RNase E binding sites and test how it affects the cleavage.
2. The manuscript is missing a more comprehensive discussion about the role of RNase E in bacteria, what do the authors consider is its role in comparison to other RNases? For example, the last sentence in the Results and Discussion section can be expanded to a paragraph.
3. The authors should consider adding another figure showing a schematic model that summarizes their findings about RNase E activity.

Minor comments:

4. Page 4, second paragraph: Adding a comment about the degree of similarity between RNase E from *E. coli* and RNase E from *R. sphaeroides* will be valuable.
5. Page 7, bottom paragraph: "it is transcribed from an own promoter..." should be changed to "it is transcribed from its own promoter"
6. Figure 1(a) and (b): The axes on the graphs are not labeled and it is not clear what their meaning is?
7. Figure 2(e) and (f): Why does the X axis reach 80 and does not stop at 10? The authors can also consider using log scale for the Y axis.
8. Figure 5: The graphs' formatting is not uniform. For example the graphs in (b) are missing a border that is present for the graphs in (a) and (c).

Reviewer #2 (Comments to the Authors (Required)):

Review Ms. Förstner et al.

The authors performed comparative RNA-Seq using a *Rhodobacter sphaeroides* wild-type strain and a derivative strain in which the endogenous *rne* gene was replaced with the temperature-sensitive *rne-3071* allele from *E. coli*. The transcriptomes were analyzed at 32°C and at the non-permissive temperature of 42°C. They observed 5'-ends that are enriched in the mutant strain as well as 5'-ends that are reduced. RNase E knockdown is shown to lead to a defect in the formation of photosynthetic complexes, resulting in retarded phototrophic growth. The study documents for an alpha-proteobacterium with versatile metabolic capacities how the activity of RNase E in RNA processing and decay moulds the bacterial transcriptome. Before publication, the manuscript requires major revision to improve its readability.

Major comments:

- 1) The manuscript needs extensive language editing; I have made numerous suggestions below.
- 2) The figures, and the lettering in particular, are too small. Use roughly equal letter size for all part figures and figures should be readable on a normal A4 printout. Also, in many figures, the panels are marked as "a", "b" etc., but are referred to in the text as panel "A", "B" etc.; please harmonize.
- 3) Some bioinformatic illustrations (Fig. 1C,D / Fig. 2 / Fig. S1) should be better explained in the figure legends and text (e.g. p.4, 3rd paragraph), such that the reader can fully understand the information that is provided; what means "base mean" (x-axis) in Fig. 1C, D? What is the reason for the morphology of the plots below base mean values of about 10? In the legend to Fig. 2, the authors should clearly state that panels A, C and E show the sum of all 5'-ends assumed to be RNase

cleavage sites (41,000), as inferred from the fact that they were depleted in the mutant at the non-permissive temperature; likewise, explain that panels B, D and F are the illustrations for the enriched sites (the 41,000 sites reduced to 23,000). The difference between panels A/C/E and B/D/F should also be immediately recognizable in the figure by differential axis lettering.

4) Fig. 1A,B: explain the boxes with light blue/cyan columns, captioned "mapped 5' ends".

5) The colour code used for the strains is suboptimal; particularly in Fig. 5C, the light green and blue shades are hard to differentiate; use colours that can be better distinguished. In Fig. 5C, move the assignment of colour (WT and mutants) to outside the box to clearly separate these dots from the data points within the box.

6) p.3, lines 22 to 26: the information on SorY is not relevant here, can be omitted.

7) p.3, line 28, make clearer what you mean here: "... mapping of RNase E cleavage site (which are reduced upon RNase E depletion) ..."

8) p.7, 2nd paragraph, discussion of data for SorX: I do not see clear coherence of NB and RNA-seq data: in the NB, pre-SorX (116 nt) is enriched in the mutant relative to WT at 32°C, but the reverse is seen for the coverage of the corresponding 5'-end; the latter increases for the mutant at 42°C vs. 32°C, but the pre-SorX signal in the NB decreases for the mutant at 42 vs 32°C. Please scrutinize the current discussion of the SorX data.

9) p.8, last sentence of the 1st paragraph appears rather unexpectedly. The authors discuss that the NB and RNA-Seq data for PcrZ do not indicate much difference between WT and mutant, and then suddenly mention that RNase E cleavage of PcrZ may have strong impact on the formation of photosynthetic complexes. Please modify.

Minor comments:

10) throughout manuscript: "Gram-negative", not "gram-negative"

11) p.3, lines 17/18: I doubt that most readers understand what is meant by an "incoherent feed-forward loop". Since the details of this loop are not important here, use a more general description, e.g. "regulatory circuit".

12) p.5, 1st paragraph, last sentence: I doubt that ref. 42 was the one that showed endonucleolytic J1 activity in *B. subtilis*.

13) p.5, 3rd paragraph, line 4: what do you mean by "within stretches of the same nucleotide"? Homonucleotide stretches? / line 5: what is meant by "in a proximity of 3 nt"? (see also p.12, line 8). Let us assume an example sequence, "5'-GCATACC" (cleavage on the 5' site of the nucleotides); when counting from the central T, do you mean to assign also cleavages up to the first G and to the last C to the same site? Please make clearer.

14) p.7, 2nd paragraph: describe first the sRNAs shown in Fig. 4 and then those documented in Fig. S3.

15) p.7, 3rd paragraph, line 4 from bottom to end of page: complete genome position "2,565" in line 2 from bottom; I also have difficulties to follow the argumentation; I see multiple 5'-ends from position 2,565,910 to 2,565,935, indicating 5'-end heterogeneity of the "55 nt" fragment ensemble; the 5'-end signals are similar for the WT and mutant at both temperatures, but the NB indicates a somewhat lower abundance of the "55 nt" fraction in the mutant. Please clarify. The authors could add a sentence stating that the NB and RNA-seq 5' end coverage data are not always consistent in terms of quantitative trends.

16) p.8, 2nd paragraph: the first sentence can be deleted.

17) p. 8, 2nd paragraph, discussion of data for RSP_7527 is rather confusing, rewrite.

18) p.12, line 4: what do you mean by "for the remaining one" ?

19) Legend to Fig. 3: what is meant by "between position -1 and 1 or 1 and 2" and by "mutant enriched sites" ?

20) Fig. S4: change y-axis lettering to "ratio OD770 to OD660" /explain abbreviations "LO" and "PT" in the figure legend

21) Fig. S6: give growth temperature in the legend

Suggestions for text changes:

p.1:

- Abstract, line 8: delete "in this GC rich alpha proteobacterium" (not relevant as abstract information)
- Introduction, line 2 rewrite to "Successful adaptation to changing conditions ..."

p.2:

- 3rd paragraph, line 5: "... by prior endonucleolytic cleavage of the RNA: Binding of RNase E to a ..."
- 4th paragraph, line 5: "...symbionts and pathogens. An important role ..."
- line 6: homeostasis
- line 8: delete comma after "mRNA"

p.3:

- line 11: "...restricted to the model organism *R. sphaeroides* whose genome was the first to become available in this group of bacteria [28,29]. RNA-seq analyses of *R. sphaeroides* were performed for the first time ..."
- line 17: "... gene expression and is part of a regulatory circuit ..."
- line 22: "... recognized (e.g., [38]). For example, in *R. sphaeroides*, ..."
- line 25: add comma after UpsM
- line 26: "Considering the acknowledged role of RNase E ..."
- line 29: "... an *rne* mutant in *R. sphaeroides*. The role of RNase E ..."
- 2nd paragraph (Results and Discussion), line 3: "...transcriptome adjustments we need ..." / line 3 from bottom: give species name (*Salmonella Typhimurium*) and reference for the study

p.4:

- line 2: "...get insight into ..."
- 2nd paragraph, line 3: "To study the relevance of RNase E cleavage for the shaping of the *R. sphaeroides* transcriptome, ..."
- 2nd paragraph, line 7: "... at 32{degree sign}C (see below), demonstrating ..."
- 2nd paragraph, line 9: "...the two strains, which were harvested in mid-exponential growth phase at 32{degree sign}C or 20 min after ..."
- 3rd paragraph, line 6, rewrite: "...shift to 42{degree sign}C, which was in most cases not observed, or to a lesser extent, with RNA derived from cultures grown at 32{degree sign}C ..."
- 3rd paragraph, line 12: comma before "may"
- last paragraph: lines 5/6: what do you mean by "(about 2 500)"?
- last paragraph: line 7: "... to 42{degree sign}C may have affected ..."

p.5:

- 3rd paragraph, line 1, rewrite: "'5' ends enriched in the wild type and those enriched in the mutant are often ..."
- last paragraph, line 7 from bottom: "Only about 2 % of the cleavage sites mapped to 3' UTRs; similar ratios were described by Chao et al. in the related study on *Salmonella* RNase E [43]".
- last paragraph, line 4 from bottom: "... percentage of enriched sites ..."

p.6:

- 3rd paragraph, sentence starting in line 3, rewrite: "Although *R. sphaeroides* has a higher genomic G,C content (~69%) than *E. coli* (~51%), *R. sphaeroides* RNase E shows a preference for A,U-rich target sites as well."
- 4th paragraph, rewrite last sentence (cleavage site rates were not measured): "While RNase E cleavage sites were observed to occur frequently in close proximity of stop codons in *Salmonella*

[43], we found a relatively high incidence of cleavage sites around start codons."

- 5th paragraph, line 2 from bottom: comma before "demonstrating"

- last paragraph, last line: comma after "processing"

p.7:

- 2nd paragraph, line 6 from bottom: "The function of this sRNA is unknown."

- 3rd paragraph, first sentence: "As SorX [34], sRNA SorY has a role in the oxidative stress response [33]. Unlike ..."

- 3rd paragraph, line 6: "Another sRNA with known function in *R. sphaeroides* is PcrZ (136 nt), which is involved in the regulation of photosynthesis gene expression [31]." (most readers will not understand what is meant by an "incoherent feed-forward loop")

- 3rd paragraph, line 6 from bottom: "... rneE.colie(ts) gene from *E. coli*. In agreement with this, the corresponding Northern blot indicates that the amount of the 136 nt transcript does not significantly vary in the two strains at the permissive or non-permissive temperature (Supplementary Figure S3D)."

p.8:

- 3rd paragraph, line 4: "... more abundant in the mutant strain at both temperatures ..."

- 3rd paragraph, last line: replace "not shown" with "Fig. S3F"

p.9:

- line 3, rephrase: "... or phototrophic growth conditions. This corresponds to the observation that phototrophic growth of the mutant is strongly retarded (Figure 5B). Remarkably, both strains showed identical growth behavior when grown chemotrophically. This finding underlines the important physiological role of RNase E in *R. sphaeroides* and further demonstrates that RNase E can have a selective impact on specific physiological processes."

- 2nd paragraph, line 3: "... (pufB and pufA for the light harvesting complex I and pufL and pufM for the reaction center), a protein involved in assembly of the complexes (pufX) and another one affecting early bacteriochlorophyll synthesis (pufQ) [52]."

- 2nd paragraph, line 7: introduce abbreviation "LHI" already in line 4

- 2nd paragraph, line 10: "Again, RNase E initiates further processing by cleaving in the 5' region of the pufL coding sequence [21,23]."

- 2nd paragraph, line 15: "However, there are more RNase E ..."

- 2nd paragraph, line 18: "... to identify primary sites that limit the overall rate of processing. As in the puf operon, ..."

- 2nd paragraph, line 21: "...light-harvesting II complex, bch genes code for enzymes of bacteriochlorophyll synthesis and crt genes for enzymes of carotenoid synthesis) ..."

- 2nd paragraph, line 25: "Some mRNAs encoding important regulators ..."

- 2nd paragraph, line 28: "Nevertheless, the strong effect ... cannot be deduced ..."

- 2nd paragraph, line 32: delete comma after "elusive"

p.10:

- line 2: delete "which are"

- line 6: "...64 and 30 per kb, respectively). All these ..."

- line 19: "However, the mutant ..."

- line 21: "This demonstrates that RNase E also affects other ..."

- line 23: "...RNase E in RNA metabolism, despite the presence of several enzymes with ribonucleolytic activity in bacteria, and ..."

- line 25: "...and decay on regulation of gene expression, an aspect that is still neglected in many studies focusing on gene regulation."

- 2nd paragraph, line 3: "...temperature-sensitive E. coli rne-3071 (ts) allele derived from strain E. coli N3431 [4,59]. Construction of this strain is described ..."
- 2nd paragraph, line 7: "...conditions in the presence of 40 W ..."
- 3rd paragraph, line 3: "...an extinction coefficient of 76 mM⁻¹ cm⁻¹ at 770 nm."

p.11:

- 1st paragraph, line 1: 1.5 ml / lines 4 and 5: units of "100 protein" and "400 ICM buffer" ?
- 2nd paragraph, line 2: diameter / line 7: "...oxidative stress agent was defined as 100 % survival. The percentage of colonies grown from the treated cultures (percent survival) was referred to the 100% of the control culture."
- 3rd paragraph, line 3: "per 1 RNA": unit lacking / same line: "...by PCR against the gloB ..."
- 3rd paragraph, line 7: "For detection of SorX, a PCR product (primers listed in Tabela S1) was labeled with ..."
- 4th paragraph, line 2: "...accessible through the GEO Series ..."

p.12:

- line 6: here the authors write "WT", throughout text they have used "wild type"; please harmonize
- line 11: what is "bedtools intersect"?
- line 11: "...annotation features ..."
- line 12: "For the detection of motifs, the sequences ..."

p.18, Figure legends:

- Fig. 1, line 12: "Global analysis of 5' end profiles at a permissive (32 {degree sign}C, panel C) and a non-permissive temperature (42 {degree sign}C, panel D)."
- Fig. 2, line 3: "...individual genes per kb sorted by gene type. (D) Box plot of enriched site density for the individual genes per kb sorted by gene type."
- Fig. 4, line 2: "For RNA isolation, strains were cultivated ..." / line 4: "5S rRNA was used ..."
- Fig. 5, line 11: "... shifted to 42 {degree sign} C for 1 h either under no-stress conditions for control or in the presence of either 0.5 mM H₂O₂, 1.5 mM tBOOH or 1 mM paraquat. For photooxidative stress, bacteria were cultivated at 32 {degree sign}C under aerobic conditions in the dark and in the presence of 0.2 μM methyleneblue until reaching an OD₆₆₀ of 0.4. Cells were then shifted to 42 {degree sign}C for 1 h ..."
- Fig. 5, line 2 from bottom: "Survival is displayed as the percentage of colony numbers forming in serial dilutions of the stressed bacterial cultures relative to the control culture (referred to as 100% survival)."

Reviewer #3 (Comments to the Authors (Required)):

The manuscript entitled "RNase E cleavage shapes the transcriptome of *Rhodobacter sphaeroides* and strongly impacts phototrophic growth" by Konrad U. Förstner and colleagues reports a transcriptome-wide mapping of RNase E cleavage sites by TIER-seq in the GC-rich alpha proteobacterium *R. sphaeroides*. As RNase E seems essential in their model organism, the authors chose to replace the endogenous *rne* gene by a temperature sensitive (*ts*) *E. coli* *rne* gene and to compare their transcriptomes at permissive and non-permissive temperatures. The relevance of the

genome-wide data produced was assessed on the processing of two UTR-derived sRNAs, UpsM [PMID: 27802301] and SorX [PMID: 27420112], known to be RNase E-dependent. More specifically, the TIER-seq strategy developed here allowed the identification of new UTR-derived sRNAs generated by RNase E processing, confirmed the key role played by RNase E on the formation of photosynthetic complexes by initiating a differential degradation of the *pufQBALMX* polycistronic mRNA in Rhodobacterales and established the role of RNase E on oxidative stress response by identifying mRNAs coding sigma factors as RNase E targets.

The manuscript falls into the scope of Life Science Alliance. The TIER-seq strategy that was implemented by the authors as developed in Chao et al. (2017) [PMID: 28061332] seems sound and provides resources that should be valuable for people working on RNA metabolism in bacteria and on metabolic switches in Rhodobacterales (e.g. stress response, phototrophic vs chemotrophic growth). Nonetheless, I have some major concerns that should be addressed by the authors before considering it for publication.

I will restrict my comments as general ones, as more specific details can be addressed in the second round of revision. Nevertheless, I advise the authors to seriously improve the writing of the manuscript before resubmission (if accepted by Life Science Alliance). The punctuation marks are mostly missing, making it difficult to read. The introduction should be restricted to what is actually discussed in the "Results and Discussion" section. Repetitions between the "Introduction" and "Results and Discussion" should be avoided when unnecessary. Some long paragraph would benefit from being sectioned. As written, the main conclusions of the section "Results and Discussion" are difficult to extract and are not put in perspective to what is known for other organisms. Many typos remain, especially in the section "Materials and Methods", should be corrected (mutant strain with two different writings, diameter with an r missing...).

My main concern is the choice of comparing transcriptome data from *R. sphaeroides* cells expressing the endogenous RNase E or an *E. coli* RNase E. As classified in Ait-Bara and Carpousis (2015) [PMID: 26096689], *E. coli* and *R. sphaeroides* RNase E are type I and type II, respectively. While type II resembles type I in the organization of their catalytic region (with the exception of an insertion in the S1 domain), type I RNase E members have a membrane targeting sequence that has not been discerned in any type II RNase E. In *E. coli* (type I) and *Caulobacter crescentus* (another alpha-proteobacterium; type II), RNase E molecules have different cellular locations that seem to result in different spatial organization of their transcriptomes [see: PMID: 27198188; PMID: 29610352; PMID: 20562858]. Therefore, one can wonder if the genome-wide data produced in this study are physiologically relevant. In my opinion, it would have been better to perform a depletion of the endogenous RNase E in the cells as performed in *Helicobacter pylori* for RNase J using a plasmid to conditionally express RNase E [PMID: 26726773]. Although the authors cannot change their data sets, it should be more extensively discussed in the manuscript. While the authors discussed how the differences in the composition of *E. coli* and *R. sphaeroides* degradosomes could impact the results, the impact of potentially different RNase E localizations is not addressed.

A second concern is an apparent confusion the authors make between RNA abundance and RNA stability as suggested by some result interpretations in the section "The effect of RNase E on maturation / stability of selected substrates" in the "Results and Discussion". While the authors clearly state in the abstract that "Bacteria adapt to changing environmental conditions by rapid changes in the transcriptome. This is achieved by adjusting rates of transcription but also by processing and degradation of RNAs.", they conclude that variation in RNA abundances on Northern blot results from a variation in stability without proving it. Indeed, at no point in the manuscript, variations in RNA transcription or RNA stability are experimentally addressed. Therefore, the title of the section should be changed and conclusion like "(...) the lack of RNase E reduces the 5' end-dependent pathway of degradation and stabilizes pre-SorX as also indicated by the Northern blot (...)" should be corrected.

Finally, the authors put emphasis that their strategy will, additionally to determining RNase E cleavage sites, give the 5' RNA ends. I am not sure to understand the difference between the two, as it is the same for me. The authors should try to better define what they imply by this distinction and draw specific conclusions. Otherwise, I would suggest to simplify their manuscript by only referring to RNase E cleavage sites.

Reviewer #1 (Comments to the Authors (Required)):

The authors globally identified the RNase E cleavage sites and 5' ends in *Rhodobacter sphaeroides* by applying the TIER-seq method. In this method the endoribonucleases are inactivated for a short time and this is followed by RNA-seq. The results demonstrate the importance of RNase E to different aspects in the life of *R. sphaeroides* such as: maturation and turn-over of RNAs or coping with oxidative stress. The paper is well-written and is easy to read. It also provides an important resource for researchers studying *Rhodobacter sphaeroides*. However, the paper could be improved by the following:

1. The authors state that "RNase E shows a preference for AU rich target sites". The only support they provide for this statement is a weak motif shown in Figure 3(a). The authors should consider additional experiments to support this statement or to better address this issue. For example, replacing the AU in one of the RNase E binding sites and test how it affects the cleavage.

Authors' response: We thank the reviewer for the suggestions. It was shown in previous publications from our group that RNase E also prefers AU-rich sequences at selected sites in *Rhodobacter capsulatus*. We added this information and the references.

2. The manuscript is missing a more comprehensive discussion about the role of RNase E in bacteria, what do the authors consider is its role in comparison to other RNases? For example, the last sentence in the Results and Discussion section can be expanded to a paragraph.

Authors' response: We added a summary figure and describe this at the end of the Results and Discussion part.

3. The authors should consider adding another figure showing a schematic model that summarizes their findings about RNase E activity.

Authors' response: We added such a figure.

Minor comments:

4. Page 4, second paragraph: Adding a comment about the degree of similarity between RNase E from *E. coli* and RNase E from *R. sphaeroides* will be valuable.

Authors' response: This information was included.

5. Page 7, bottom paragraph: "it is transcribed from an own promoter..." should be changed to "it is transcribed from its own promoter"

Authors' response: The text was changed.

6. Figure 1(a) and (b): The axes on the graphs are not labeled and it is not clear what their meaning is?

Authors' response: We have modified the figure accordingly.

7. Figure 2(e) and (f): Why does the X axis reach 80 and does not stop at 10? The authors can also consider using log scale for the Y axis.

Authors' response: The 80 were chosen as there are genes with up to 80 sites but in very low frequency. We have adapted the image and the range goes now up to 40 sites per gene while keeping as zoomed extract to the frequencies of 1 - 10 sites.

8. Figure 5: The graphs' formatting is not uniform. For example the graphs in (b) are missing a boarder that is present for the graphs in (a) and (c).

Authors' response: We understand the potential disbalance that the reviewer sees here intuitively. But all panels have a grid in the actual graph canvas. Adding an additional border around (B) would be non-functional visual noise that distracts from the actual information (see Tufte, Edward R. (1983). *The Visual Display of Quantitative Information*. Cheshire).

Reviewer #2 (Comments to the Authors (Required)):

Review Ms. Förstner et al.

The authors performed comparative RNA-Seq using a *Rhodobacter sphaeroides* wild-type strain and a derivative strain in which the endogenous *rne* gene was replaced with the temperature-sensitive *rne-3071* allele from *E. coli*. The transcriptomes were analyzed at 32{degree sign}C and at the non-permissive temperature of 42{degree sign}C. They observed 5'-ends that are enriched in the mutant strain as well as 5'-ends that are reduced. RNase E knockdown is shown to lead to a defect in the formation of photosynthetic complexes, resulting in retarded phototrophic growth. The study documents for an alpha-proteobacterium with versatile metabolic capacities how the activity of RNase E in RNA processing and decay moulds the bacterial transcriptome. Before publication, the manuscript requires major revision to improve its readability.

Major comments:

1) The manuscript needs extensive language editing; I have made numerous suggestions below.

Authors' response: We are thankful for the detailed and helpful comments and changed the text accordingly.

2) The figures, and the lettering in particular, are too small. Use roughly equal letter size for all part figures and figures should be readable on a normal A4 printout. Also, in many figures, the panels are marked as "a", "b" etc., but are referred to in the text as panel "A", "B" etc.; please harmonize.

Authors' response: We thank reviewer for pointing to this and have addressed this in the whole manuscript as well as the supplementary material.

3) Some bioinformatic illustrations (Fig. 1C,D / Fig. 2 / Fig. S1) should be better explained in the figure legends and text (e.g. p.4, 3rd paragraph), such that the reader can fully understand the information that is provided; what means "base mean" (x-axis) in Fig. 1C, D? What is the reason

for the morphology of the plots below base mean values of about 10? In the legend to Fig. 2, the authors should clearly state that panels A, C and E show the sum of all 5'-ends assumed to be RNase cleavage sites (41,000), as inferred from the fact that they were depleted in the mutant at the non-permissive temperature; likewise, explain that panels B, D and F are the illustrations for the enriched sites (the 41,000 sites reduced to 23,000). The difference between panels A/C/E and B/D/F should also be immediately recognizable in the figure by differential axis lettering.

Authors' response: The morphology of the plots at low coverage levels is partially a result of the fold-change shrinkage performed by DESeq2 that prevents high fold-changes which occur due to the stronger impact of noise at these lower values (see Love *et al.*, Genome Biology, 2014). We have extended the figure legend to explain the data.

4) Fig. 1A,B: explain the boxes with light blue/cyan columns, captioned "mapped 5' ends".

Authors' response: We have added a detailed description of these MA plots.

5) The colour code used for the strains is suboptimal; particularly in Fig. 5C, the light green and blue shades are hard to differentiate; use colours that can be better distinguished. In Fig. 5C, move the assignment of colour (WT and rne ts) to outside the box to clearly separate these dots from the data points within the box.

Authors' response: We thank the reviewer for the feedback and have changed the figure to a color scheme with stronger contrast based on it.

6) p.3, lines 22 to 26: the information on SorY is not relevant here, can be omitted.

Authors' response: At that place we state that RNA processing and degradation is important for the generation / function of sRNAs in *R. sphaeroides*. We thank the reviewer for the constructive feedback but since SorY alters the stability of its target it should be mentioned here.

7) p.3, line 28, make clearer what you mean here: "... mapping of RNase E cleavage site (which are reduced upon RNase E depletion) ..."

Authors' response: We modified this as suggested.

8) p.7, 2nd paragraph, discussion of data for SorX: I do not see clear coherence of NB and RNA-seq data: in the NB, pre-SorX (116 nt) is enriched in the mutant relative to WT at 32°C, but the reverse is seen for the coverage of the corresponding 5'-end; the latter increases for the mutant at 42°C vs. 32°C, but the pre-SorX signal in the NB decreases for the mutant at 42 vs 32°C. Please scrutinize the current discussion of the SorX data.

Authors' response: We modified this paragraph. A band on a Northern blot represents a certain RNA species, not a 5' end. If there is less processing at the downstream site, more pre-SorX accumulates. The 5' end which are mapped can stem from the 116 nt pre SorX or from the 5' processing product.

9) p.8, last sentence of the 1st paragraph appears rather unexpectedly. The authors discuss

that the NB and RNA-Seq data for PcrZ do not indicate much difference between WT and mutant, and then suddenly mention that RNase E cleavage of PcrZ may have strong impact on the formation of photosynthetic complexes. Please modify.

Authors' response: We modified this paragraph.

Minor comments:

10) throughout manuscript: "Gram-negative", not "gram-negative"

Authors' response: This was corrected.

11) p.3, lines 17/18: I doubt that most readers understand what is meant by an "incoherent feed-forward loop". Since the details of this loop are not important here, use a more general description, e.g. "regulatory circuit".

Authors' response: We changed this.

12) p.5, 1st paragraph, last sentence: I doubt that ref. 42 was the one that showed endonucleolytic J1 activity in *B. subtilis*.

Authors' response: We agree that this reference describes the RNase J activity in *Rhodobacter* and added another reference for RNase J in *B. subtilis*.

13) p.5, 3rd paragraph, line 4: what do you mean by "within stretches of the same nucleotide"? Homonucleotide stretches? / line 5: what is meant by "in a proximity of 3 nt"? (see also p.12, line 8). Let us assume an example sequence, "5'-GCATACC" (cleavage on the 5' site of the nucleotides); when counting from the central T, do you mean to assign also cleavages up to the first G and to the last C to the same site? Please make clearer.

Authors' response: We changed the text accordingly and added a link to the clustering tool.

14) p.7, 2nd paragraph: describe first the sRNAs shown in Fig. 4 and then those documented in Fig. S3.

Authors' response: We find it more appropriate to describe sRNAs which stem from similar processing events (e.g. processing from the 5' UTR) together and not to switch back and forth between those types.

15) p.7, 3rd paragraph, line 4 from bottom to end of page: complete genome position "2,565" in line 2 from bottom;

Authors' response: Please excuse this mistake. This should read 2,565,914 and was corrected.

I also have difficulties to follow the argumentation; I see multiple 5'-ends from position 2,565,910 to 2,565,935, indicating 5'-end heterogeneity of the "55 nt" fragment ensemble; the 5'-end signals are similar for the WT and mutant at both temperatures, but the NB indicates a somewhat lower abundance of the "55 nt" fraction in the mutant. Please clarify.

Authors' response: Most of the 5' ends mapping to this region are similar in the two strains, while the 5' end at position 2,565,914 is clearly more abundant in the WT versus the mutant strain and also much more abundant than the other 5' ends.

The authors could add a sentence stating that the NB and RNA-seq 5' end coverage data are not always consistent in terms of quantitative trends.

Authors' response: This is certainly true. However in this case the decreased peak in the mutant represents less cleavage at this position, which would reduce the band stemming from processing.

16) p.8, 2nd paragraph: the first sentence can be deleted.

Authors' response: We personally would prefer to keep this sentence in order to keep the context.

17) p. 8, 2nd paragraph, discussion of data for RSP_7527 is rather confusing, rewrite.

Authors' response: We modified the text.

18) p.12, line 4: what do you mean by "for the remaining one" ?

Authors' response: This was improved by changing the sentence to "The first-base-in-read-coverage values of the remaining nucleotide positions were used as input for enrichment analysis with DESeq2 (version 1.16.1, Love et al, 2014)."

19) Legend to Fig. 3: what is meant by "between position -1 and 1 or 1 and 2" and by "mutant enriched sites" ?

Authors' response: We modified the text.

20) Fig. S4: change y-axis lettering to "ratio OD770 to OD660" /explain abbreviations "LO" and "PT" in the figure legend

Authors' response: The figure was adapted accordingly.

21) Fig. S6: give growth temperature in the legend

Authors' response: We added the information.

Suggestions for text changes:

P.1:

- Abstract, line 8: delete "in this GC rich alpha proteobacterium" (not relevant as abstract information)

Authors' response: We think that it is important to note that this study was performed with a bacterium, which is in contrast to *E. coli* GC-rich and would like to keep this information. Especially for the discussion of the sequence motif of the cleavage sites this is of relevance.

- Introduction, line 2 rewrite to "Successful adaptation to changing conditions ..."

Authors' response: We modified the text.

p.2:

- 3rd paragraph, line 5: "... by prior endonucleolytic cleavage of the RNA: Binding of RNase E to a ..."

- 4th paragraph, line 5: "...symbionts and pathogens. An important role ..."

- line 6: homeostasis

- line 8: delete comma after "mRNA"

Authors' response: All corrections were performed.

p.3:

- line 11: "...restricted to the model organism R. sphaeroides whose genome was the first to become available in this group of bacteria [28,29]. RNA-seq analyses of R. sphaeroides were performed for the first time ..."

- line 17: "... gene expression and is part of a regulatory circuit ..."

- line 22: "... recognized (e.g., [38]). For example, in R. sphaeroides, ..."

- line 25: add comma after UpsM

- line 26: "Considering the acknowledged role of RNase E ..."

- line 29: "... an rne mutant in R. sphaeroides. The role of RNase E ..."

- 2nd paragraph (Results and Discussion), line 3: "...transcriptome adjustments we need ..." / line 3 from bottom: give species name (Salmonella Typhimurium) and reference for the study

Authors' response: All corrections were performed.

p.4:

- line 2: "...get insight into ..."

- 2nd paragraph, line 3: "To study the relevance of RNase E cleavage for the shaping of the R. sphaeroides transcriptome, ..."

- 2nd paragraph, line 7: "... at 32{degree sign}C (see below), demonstrating ..."

- 2nd paragraph, line 9: "...the two strains, which were harvested in mid-exponential growth phase at 32{degree sign}C or 20 min after ..."

- 3rd paragraph, line 6, rewrite: "...shift to 42{degree sign}C, which was in most cases not observed, or to a lesser extent, with RNA derived from cultures grown at 32{degree sign}C ..."

- 3rd paragraph, line 12: comma before "may"

- last paragraph: lines 5/6: what do you mean by "(about 2 500)"?

- last paragraph: line 7: "... to 42{degree sign}C may have affected ..."

Authors' response: All changes were made.

p.5:

- 3rd paragraph, line 1, rewrite: "5' ends enriched in the wild type and those enriched in the mutant are often ..."

- last paragraph, line 7 from bottom: "Only about 2 % of the cleavage sites mapped to 3' UTRs; similar ratios were described by Chao et al. in the related study on Salmonella RNase E [43]".

- last paragraph, line 4 from bottom: "... percentage of enriched sites ..."

Authors' response: All changes were made.

p.6:

- 3rd paragraph, sentence starting in line 3, rewrite: "Although *R. sphaeroides* has a higher genomic G,C content (~69%) than *E. coli* (~51%), *R. sphaeroides* RNase E shows a preference for A,U-rich target sites as well."

Authors' response: this part was rewritten to include more information.

- 4th paragraph, rewrite last sentence (cleavage site rates were not measured): "While RNase E cleavage sites were observed to occur frequently in close proximity of stop codons in *Salmonella* [43], we found a relatively high incidence of cleavage sites around start codons."
- 5th paragraph, line 2 from bottom: comma before "demonstrating"
- last paragraph, last line: comma after "processing"

Authors' response: All changes were made.

p.7:

- 2nd paragraph, line 6 from bottom: "The function of this sRNA is unknown."
- 3rd paragraph, first sentence: "As SorX [34], sRNA SorY has a role in the oxidative stress response [33]. Unlike ..."
- 3rd paragraph, line 6: "Another sRNA with known function in *R. sphaeroides* is PcrZ (136 nt), which is involved in the regulation of photosynthesis gene expression [31]." (most readers will not understand what is meant by an "incoherent feed-forward loop")
- 3rd paragraph, line 6 from bottom: "... rneE.colie(ts) gene from *E. coli*. In agreement with this, the corresponding Northern blot indicates that the amount of the 136 nt transcript does not significantly vary in the two strains at the permissive or non-permissive temperature (Supplementary Figure S3D)."

Authors' response: all changes were made.

p.8:

- 3rd paragraph, line 4: "... more abundant in the mutant strain at both temperatures ..."
- 3rd paragraph, last line: replace "not shown" with "Fig. S3F"

Authors' response: all changes were made.

p.9:

- line 3, rephrase: "... or phototrophic growth conditions. This corresponds to the observation that phototrophic growth of the mutant is strongly retarded (Figure 5B). Remarkably, both strains showed identical growth behavior when grown chemotrophically. This finding underlines the important physiological role of RNase E in *R. sphaeroides* and further demonstrates that RNase E can have a selective impact on specific physiological processes."
- 2nd paragraph, line 3: "... (pufB and pufA for the light harvesting complex I and pufL and pufM for the reaction center), a protein involved in assembly of the complexes (pufX) and another one affecting early bacteriochlorophyll synthesis (pufQ) [52]."
- 2nd paragraph, line 7: introduce abbreviation "LHI" already in line 4
- 2nd paragraph, line 10: "Again, RNase E initiates further processing by cleaving in the 5' region of the pufL coding sequence [21,23]."

- 2nd paragraph, line 15: "However, there are more RNase E ..."
- 2nd paragraph, line 18: "... to identify primary sites that limit the overall rate of processing. As in the puf operon, ..."
- 2nd paragraph, line 21: "...light-harvesting II complex, bch genes code for enzymes of bacteriochlorophyll synthesis and crt genes for enzymes of carotenoid synthesis) ..."
- 2nd paragraph, line 25: " Some mRNAs encoding important regulators ..."
- 2nd paragraph, line 28: "Nevertheless, the strong effect ... cannot be deduced ..."
- 2nd paragraph, line 32: delete comma after "elusive"

Authors' response: All changes were made.

p.10:

- line 2: delete "which are"
- line 6: "...64 and 30 per kb, respectively). All these ..."
- line 19: "However, the mutant ..."
- line 21: "This demonstrates that RNase E also affects other ..."
- line 23: "...RNase E in RNA metabolism, despite the presence of several enzymes with ribonucleolytic activity in bacteria, and ..."
- line 25: "...and decay on regulation of gene expression, an aspect that is still neglected in many studies focusing on gene regulation."
- 2nd paragraph, line 3: "...temperature-sensitive E. coli rne-3071 (ts) allele derived from strain E. coli N3431 [4,59]. Construction of this strain is described ..."
- 2nd paragraph, line 7: "...conditions in the presence of 40 W ..."
- 3rd paragraph, line 3: "...an extinction coefficient of 76 mM⁻¹ cm⁻¹ at 770 nm."

Authors' response: all changes were made.

p.11:

- 1st paragraph, line 1: 1.5 ml / lines 4 and 5: units of "100 protein" and "400 ICM buffer" ?
- 2nd paragraph, line 2: diameter / line 7: "...oxidative stress agent was defined as 100 % survival. The percentage of colonies grown from the treated cultures (percent survival) was referred to the 100% of the control culture."
- 3rd paragraph, line 3: "per 1 RNA": unit lacking / same line: "...by PCR against the gloB ..."
- 3rd paragraph, line 7: "For detection of SorX, a PCR product (primers listed in Tabel S1) was labeled with ..."
- 4th paragraph, line 2: "...accessible through the GEO Series ..."

Authors' response: All changes were made.

p.12:

- line 6: here the authors write "WT", throughout text they have used "wild type"; please harmonize
- line 11: what is "bedtools intersect"?
- line 11: "...annotation features ..."
- line 12: "For the detection of motifs, the sequences ..."

Authors' response: All changes were made.

p.18, Figure legends:

- Fig. 1, line 12: "Global analysis of 5' end profiles at a permissive (32 {degree sign}C, panel C) and a non-permissive temperature (42 {degree sign}C, panel D)."
- Fig. 2, line 3: "...individual genes per kb sorted by gene type. (D) Box plot of enriched site density for the individual genes per kb sorted by gene type."
- Fig. 4, line 2: "For RNA isolation, strains were cultivated ..." / line 4: "5S rRNA was used ..."
- Fig. 5, line 11: "... shifted to 42 {degree sign} C for 1 h either under no-stress conditions for control or in the presence of either 0.5 mM H₂O₂, 1.5 mM tBOOH or 1 mM paraquat. For photooxidative stress, bacteria were cultivated at 32 {degree sign}C under aerobic conditions in the dark and in the presence of 0.2 μM methyleneblue until reaching an OD₆₆₀ of 0.4. Cells were then shifted to 42 {degree sign}C for 1 h ..."
- Fig. 5, line 2 from bottom: "Survival is displayed as the percentage of colony numbers forming in serial dilutions of the stressed bacterial cultures relative to the control culture (referred to as 100% survival)."

Authors' response: All changes were made.

Reviewer #3 (Comments to the Authors (Required)):

The manuscript entitled "RNase E cleavage shapes the transcriptome of *Rhodobacter sphaeroides* and strongly impacts phototrophic growth" by Konrad U. Förstner and colleagues reports a transcriptome-wide mapping of RNase E cleavage sites by TIER-seq in the GC-rich alpha proteobacterium *R. sphaeroides*. As RNase E seems essential in their model organism, the authors chose to replace the endogenous *rne* gene by a temperature sensitive (*ts*) *E. coli* *rne* gene and to compare their transcriptomes at permissive and non-permissive temperatures. The relevance of the genome-wide data produced was assessed on the processing of two UTR-derived sRNAs, UpsM [PMID: 27802301] and SorX [PMID: 27420112], known to be RNase E-dependent. More specifically, the TIER-seq strategy developed here allowed the identification of new UTR-derived sRNAs generated by RNase E processing, confirmed the key role played by RNase E on the formation of photosynthetic complexes by initiating a differential degradation of the *pufQBALMX* polycistronic mRNA in Rhodobacterales and established the role of RNase E on oxidative stress response by identifying mRNAs coding sigma factors as RNase E targets. The manuscript falls into the scope of Life Science Alliance. The TIER-seq strategy that was implemented by the authors as developed in Chao et al. (2017) [PMID: 28061332] seems sound and provides resources that should be valuable for people working on RNA metabolism in bacteria and on metabolic switches in Rhodobacterales (e.g. stress response, phototrophic vs chemotrophic growth). Nonetheless, I have some major concerns that should be addressed by the authors before considering it for publication.

I will restrict my comments as general ones, as more specific details can be addressed in the second round of revision. Nevertheless, I advise the authors to seriously improve the writing of the manuscript before resubmission (if accepted by Life Science Alliance). The punctuation

marks are mostly missing, making it difficult to read. The introduction should be restricted to what is actually discussed in the "Results and Discussion" section. Repetitions between the "Introduction" and "Results and Discussion" should be avoided when unnecessary. Some long paragraph would benefit from being sectioned. As written, the main conclusions of the section "Results and Discussion" are difficult to extract and are not put in perspective to what is known for other organisms. Many typos remain, especially in the section "Materials and Methods", should be corrected (mutant strain with two different writings, diameter with an r missing...). My main concern is the choice of comparing transcriptome data from *R. sphaeroides* cells expressing the endogenous RNase E or an *E. coli* ts RNase E. As classified in Ait-Bara and Carpousis (2015) [PMID: 26096689], *E. coli* and *R. sphaeroides* RNase E are type I and type II, respectively. While type II resembles type I in the organization of their catalytic region (with the exception of an insertion in the S1 domain), type I RNase E members have a membrane targeting sequence that has not been discerned in any type II RNase E. In *E. coli* (type I) and *Caulobacter crescentus* (another alpha-proteobacterium; type II), RNase E molecules have different cellular locations that seem to result in different spatial organization of their transcriptomes [see: PMID: 27198188; PMID: 29610352; PMID: 20562858]. Therefore, one can wonder if the genome-wide data produced in this study are physiologically relevant. In my opinion, it would have been better to perform a depletion of the endogenous RNase E in the cells as performed in *Helicobacter pylori* for RNase J using a plasmid to conditionally express RNase E [PMID: 26726773]. Although the authors cannot change their data sets, it should be more extensively discussed in the manuscript. While the authors discussed how the differences in the composition of *E. coli* and *R. sphaeroides* degradosomes could impact the results, the impact of potentially different RNase E localizations is not addressed.

Authors' response: We agree on the comments of possible different localization of the enzymes and have discussed this as suggested by the referee. For this study it was important to have a strain with lower RNase E activity and this is well achieved by our construct.

A second concern is an apparent confusion the authors make between RNA abundance and RNA stability as suggested by some result interpretations in the section "The effect of RNase E on maturation / stability of selected substrates" in the "Results and Discussion". While the authors clearly state in the abstract that "Bacteria adapt to changing environmental conditions by rapid changes in the transcriptome. This is achieved by adjusting rates of transcription but also by processing and degradation of RNAs.", they conclude that variation in RNA abundances on Northern blot results from a variation in stability without proving it. Indeed, at no point in the manuscript, variations in RNA transcription or RNA stability are experimentally addressed. Therefore, the title of the section should be changed and conclusion like "(...) the lack of RNase E reduces the 5' end-dependent pathway of degradation and stabilizes pre-SorX as also indicated by the Northern blot (...)" should be corrected.

Authors' response: We changed the title of this section. The statements on the involvement of the 5' end-dependent pathway are based on the abundance of 5' ends. We tried to be more careful with our wording.

Finally, the authors put emphasis that their strategy will, additionally to determining RNase E cleavage sites, give the 5' RNA ends. I am not sure to understand the difference between the

two, as it is the same for me. The authors should try to better define what they imply by this distinction and draw specific conclusions. Otherwise, I would suggest to simply their manuscript by only referring to RNase E cleavage sites.

Authors' response: We are not sure, what the reviewer is questioning here. Mapping of the 5' ends was used to identify two different type of sites: Type 1: 5' ends that are present in the wild type but reduced in the mutant stem from RNase E cleavage as shown in Fig. 1 (A). Since such 5' ends often occur at adjacent nucleotides, we merge such adjacent 5' ends to a single cleavage site to get a better view on the number and distribution of cleavage sites. This takes into account that RNase E may not exclusively cut at a single position within sequence with mononucleotide stretches.

Type 2: 5' ends that are more abundant in the mutant were designated "enriched sites". They can be attributed to RNAs, which are degraded by the 5' end dependent pathway as shown in Fig. 1 (B). The abstract clearly states these two types: "We applied TIER-seq (transiently inactivating an endoribonuclease followed by RNA-seq) for the transcriptome-wide identification of RNase E cleavage sites and of 5' RNA ends, which are enriched when RNase E activity is reduced in *Rhodobacter sphaeroides*."

2nd Revision - Editorial Decision: July 10, 2018 July 10, 2018

RE: Life Science Alliance Manuscript #LSA-2018-00080-TR

Dr. Konrad Ulrich Förstner
Core Unit Systems Medicine, Universität Würzburg
Josef-Schneider-Str. 2/D15
Wuerzburg D-97080
Germany

Dear Dr. Förstner,

Thank you for submitting your revised manuscript entitled "RNase E cleavage shapes the transcriptome of *R. sphaeroides* and strongly impacts phototrophic growth". I appreciate the way you addressed the reviewers comments, and would thus be happy to publish your paper in Life Science Alliance pending final revisions necessary to meet our formatting guidelines.

Please make sure to mention everywhere the number of replicates analyzed, and please add callouts to panels E and F for Fig2. Please also provide the source data for all northern blots performed.

A. FINAL FILES:

-- High-resolution figure, supplementary figure and video files uploaded as individual files: See our detailed guidelines for preparing your production-ready images, <http://life-science-alliance.org/authorguide>

B. MANUSCRIPT ORGANIZATION AND FORMATTING:

Full guidelines are available on our Instructions for Authors page, <http://life-science-alliance.org/authorguide>

Sincerely,

3rd Revision - Editorial Decision: July 13, 2018 July 13, 2018

RE: Life Science Alliance Manuscript #LSA-2018-00080-TRR

Dr. Konrad Ulrich Förstner
Core Unit Systems Medicine, Universität Würzburg
Josef-Schneider-Str. 2/D15
Wuerzburg D-97080
Germany

Dear Dr. Förstner,

Thank you for submitting your Research Article entitled "RNase E cleavage shapes the transcriptome of *R. sphaeroides* and strongly impacts phototrophic growth". It is a pleasure to let you know that your manuscript is now accepted for publication in Life Science Alliance. Congratulations on this interesting work.

The final published version of your manuscript will be deposited by us to PubMed Central (PMC) as soon as we are allowed to do so, the application for PMC indexing has been filed. You may be eligible to also deposit your Life Science Alliance article in PMC or PMC Europe yourself, which will then allow others to find out about your work by Pubmed searches right away. Such author-initiated deposition is possible/mandated for work funded by eg NIH, HHMI, ERC, MRC, Cancer Research UK, Telethon, EMBL.

Please also see:

<https://www.ncbi.nlm.nih.gov/pmc/about/authorms/>

<https://europepmc.org/Help#howsubsmanu>

DISTRIBUTION OF MATERIALS:

Again, congratulations on a very nice paper. I hope you found the review process to be constructive and are pleased with how the manuscript was handled editorially. We look forward to future exciting submissions from your lab.

Sincerely,
